# Real-time imaging of Huntingtin aggregates diverting target search and gene transcription

**Li Li[1,2], Hui Liu[1], Peng Dong[1], Dong Li[1], Wesley R Legant[1], Jonathan B Grimm[1], Luke D Lavis[1,3], Eric Betzig[1], Robert Tjian[1,2,3*], Zhe Liu[1,3*]**

[1]Janelia Research Campus, Howard Hughes Medical Institute, Ashburn, United States; [2]LKS Bio-medical and Health Sciences Center, CIRM Center of Excellence, University of California, Berkeley, United States; [3]Transcription Imaging Consortium, Howard Hughes Medical Institute, Ashburn, United States

**Abstract** The presumptive altered dynamics of transient molecular interactions in vivo contributing to neurodegenerative diseases have remained elusive. Here, using single-molecule localization microscopy, we show that disease-inducing Huntingtin (mHtt) protein fragments display three distinct dynamic states in living cells – 1) fast diffusion, 2) dynamic clustering and 3) stable aggregation. Large, stable aggregates of mHtt exclude chromatin and form 'sticky' decoy traps that impede target search processes of key regulators involved in neurological disorders. Functional domain mapping based on super-resolution imaging reveals an unexpected role of aromatic amino acids in promoting protein-mHtt aggregate interactions. Genome-wide expression analysis and numerical simulation experiments suggest mHtt aggregates reduce transcription factor target site sampling frequency and impair critical gene expression programs in striatal neurons. Together, our results provide insights into how mHtt dynamically forms aggregates and disrupts the finely-balanced gene control mechanisms in neuronal cells.

**\*For correspondence:** jmlim@ berkeley.edu (RT); liuz11@janelia. hhmi.org (ZL)

## Introduction

Poly-glutamine (PolyQ) expansion in proteins is associated with multiple neuro- and muscle- degenerative diseases such as Huntington's disease (HD), X-linked spinobulbar muscular atrophy (SBMA) and various spinocerebellar ataxias (SCA1, 2, 3, 6, 7 and 17) (reviewed in [*Blum et al., 2013*; *Mohan et al., 2014*]). In HD, a single mutant *Huntingtin* allele with PolyQ tracts greater than 37 glutamines leads to selective cell death in the striatum and certain regions of the cortex, causing muscle coordination and cognitive defects (*Group, 1993*; *Ross and Tabrizi, 2011*). It has been widely observed that extended PolyQ tracts facilitate the formation of protein aggregates in the cytoplasm and nucleus of diseased cells (*Bates, 2003*; *DiFiglia et al., 1997*; *Huang et al., 2015*). Previous FRAP, FCS and in vitro super-resolution imaging provides significant insights into mHtt aggregate formation (*Cheng et al., 2013*; *Duim et al., 2014*; *Kim et al., 2002*; *Park et al., 2012*; *Sahl et al., 2012*; *Wustner et al., 2012*). However, the dynamics of aggregate formation or how the resulting 'plaques' might influence essential molecular transactions that disrupt gene expression programs have not been investigated at the single-molecule level in living cells.

Since the original discovery of mHtt aggregates in the nucleus and cytoplasm of HD cells, the relevance of these aggregates or plaques to disease pathology has been under vigorous debate (*DiFiglia et al., 1997*; *Scherzinger et al., 1997*; *Woerner et al., 2016*). Currently, several mechanisms have been proposed to explain how mHtt aggregates might contribute to disease states. Interestingly, it was shown that the formation of PolyQ aggregates can in some instances, protect

**eLife digest** Huntington's disease belongs to a group of human genetic disorders in which faulty proteins cause nerve cells to progressively die. All proteins are made from building blocks called amino acids, and these diseases are collectively called PolyQ expansion diseases because the faulty proteins usually have an abnormally long stretch that contains many repeats of an amino acid called glutamine. The stretches of glutamines make these mutant proteins stick to each other, which means that they aggregate in the diseased cells.

Researchers have proposed several mechanisms to explain how aggregates of one such mutant protein, called mutant Huntingtin (mHtt), might contribute to Huntington's disease. One possibility is that mHtt accumulates in the nucleus of the cell – which houses most of the cell's DNA – and interferes with the proteins that are required to switch genes on or off. Preventing these gene regulatory proteins from carrying out their role could disrupt the normal pattern of gene activity. However, working out if this is the case would require researchers being able to follow how mHtt forms aggregates and interferes with normal processes in living cells.

Li et al. have now used high-resolution, high-speed microscopy to directly track the movement of individual mHtt proteins in living mouse cells in real time. The analysis showed that PolyQ protein fragments of mHtt behaved in three distinct ways – they diffused, briefly clustered or stably aggregated. Large stable aggregates of mHtt created decoy-like traps that interfered with the behavior of the gene regulatory proteins. Using computer-aided simulations, Li et al. then showed that that these molecular traps make the proteins take much longer to find their true target genes. Further studies in nerve cells suggested that this phenomenon disrupted the normal pattern of gene activity.

Next, Li et al. also used the live cell imaging system to look at which gene regulatory proteins were sequestered in the mHtt aggregates. This approach identified proteins that are important regulators involved in a number of neurological disorders. These proteins included some with long stretches of glutamines and unexpectedly others in which the glutamines were interspersed with other amino acids. Aggregates of mHtt therefore appear to affect a much broader range of proteins than previously thought. Together, the results provide insight into how mHtt forms aggregates and disrupts the finely balanced mechanisms that control gene activity in nerve cells. Future studies could explore the general principles that determine which proteins interact with mHtt aggregates. This could help reveal the faulty processes that underlie Huntington's disease and other neurodegenerative disorders.

cells from apoptosis in short-term cell culture experiments (*Saudou et al., 1998*; *Taylor et al., 2003*). Specifically, it was proposed that soluble fragments or oligomers of mHtt are more toxic than mHtt aggregates. Stable self-aggregation of mHtt monomers was postulated to neutralize prion protein interacting surfaces and protect cells from prion induced damage (*Arrasate et al., 2004*; *Saudou et al., 1998*; *Slow et al., 2005*). However, this model does not address the long-term effect of mhtt aggregates in striatal cells nor does it exonerate mHtt aggregates from potentially contributing to the disease state. For example, myriad studies have reported the toxicity of aggregates in vivo (*Labbadia and Morimoto, 2013*; *Michalik and Van Broeckhoven, 2003*; *Williams and Paulson, 2008*; *Woerner et al., 2016*). Without methods to directly observe and measure biochemical reactions and molecular interactions in living cells, it is challenging to gain mechanistic insights that may help resolve these controversies.

With recent advances in imaging and chemical dye development (reviewed in [*Liu et al., 2015*]), it has become possible to detect and track individual protein molecules in single living cells (*Abrahamsson et al., 2013*; *Chen et al., 2014a*, *2014b*; *Elf et al., 2007*; *Gebhardt et al., 2013*; *Grimm et al., 2015*; *Hager et al., 2009*; *Izeddin et al., 2014*; *Liu et al., 2014*; *Mazza et al., 2012*; *Mueller et al., 2013*). Decoding the complex behavior of single molecules enables us to measure molecular kinetics at a fundamental level that is often obscured in ensemble experiments. Specifically, the rapidly emerging high-resolution fast image acquisition platforms provide a means for visualizing and measuring the in vivo behavior of dynamically regulated molecular binding events. It also

becomes possible to generate 3D molecular interaction maps in living mammalian cells and elucidate local diffusion patterns in the highly heterogeneous sub-cellular environment (*Chen et al., 2014a*, *2014b*; *Izeddin et al., 2014*; *Liu et al., 2014*).

Here, using HD as the model, we devised a molecular imaging system to quantify the formation of protein structures and measure the real-time dynamics and behavior of PolyQ-rich proteins. First, with live-cell PALM and FRAP experiments, we compared gross structures and diffusion dynamics of wild-type (Htt-25Q) versus disease-inducing mutant (mHtt-94Q) Htt protein fragments. Interestingly, soluble fractions of wild-type Htt-25Q and mutant Htt-94Q display similar rapid diffusion kinetics. Strikingly, both Htt-25Q and mHtt-94Q protein fragments also form small, diffraction-limited clusters in live cells. These clusters are highly dynamic and resolve quickly (mean lifetime < 10~20 s). However, the mutant Htt-94Q protein also forms much larger highly stable aggregates (FRAP recovery lifetime > 45 min). Two-color super resolution imaging reveals that mHtt aggregates exclude chromatin and selectively interact with a set of neurological disease-related factors (Foxp2, TBP, Sp1 *and* wild-type Huntingtin). Fine domain mapping experiments suggest that continuous PolyQ tracts contribute to but are not necessary for binding to mHtt aggregates. Notably, we found that aromatic amino acids enabled proteins with sparse glutamines to bind mHtt aggregates. Single-molecule tracking and numerical simulation experiments suggest that mHtt aggregates act as large decoy traps in the cell, efficiently slowing down target search kinetics and decreasing target site sampling frequencies. Consistent with this model, unbiased genomic screens showed that elevating Sp1 levels in HD-affected striatal cells partially restored the expression of Sp1 dependent target genes. Sp1 target site occupancies are also significantly decreased in HD-affected cells. These findings provide new insights into potential Huntington's disease mechanisms and reveal the role of mHtt aggregates in disrupting key dynamic biological processes in living cells.

## Results

### Live-cell Htt dynamics

To establish a cellular imaging system for HD and evaluate the effects of PolyQ expansion on Htt protein dynamics, we fused disease-inducing Huntingtin protein exon 1 fragment (mHtt-94Q) (*Rubinsztein, 2002*; *Yamamoto et al., 2000*) and its wild-type counterpart (Htt-25Q) (*Krobitsch and Lindquist, 2000*) to a monomeric photo-switchable protein (mEOS3.2 (*Zhang et al., 2012*); w/o nuclear localization signal - NLS). We stably expressed each fragment in mouse embryonic stem (ES) cells or STH*dh* striatal cells (*Trettel et al., 2000*). Epi-fluorescence live imaging revealed that 8~15% of the mHtt-94Q cells (27 of 234 cells) contain large visible sub-cellular aggregates while the majority (123 of 125 cells) of Htt-25Q cells show a relatively homogeneous intracellular distribution (*Figure 1—figure supplement 1A*).

To measure Htt protein dynamics, we next performed live-cell imaging experiments by stochastically switching the mEOS3.2 moiety to the red-shifted state and tracking single-molecule movement of the activated molecules (sptPALM) (*Manley et al., 2008*). This technique allowed us to generate high-density single-molecule trajectories separated by stochastic activation times. As expected, ensemble whole-cell diffusion and velocity analysis reveals that a much larger fraction of mHtt-94Q molecules show slower and more compact diffusion kinetics compared to Htt-25Q controls (*Figure 1B*, *Figure 1—figure supplement 1B and C*), suggesting that mHtt-94Q is on average significantly less mobile. High-density trajectories generated with sptPALM also allows reconstruction of high-resolution sub-cellular diffusion maps (*Figure 1A*, middle and bottom panels). Interestingly, both mean square displacement (MSD) and Bayesian inference-based diffusion maps show that the slow mHtt-94Q species were mainly concentrated around aggregates, while mHtt-94Q molecules in other regions remain fast moving. Consistent with these observations, fluorescence recovery after photo-bleaching (FRAP) experiments within non-aggregate containing regions show full recoveries within 1 s (*Figure 1—figure supplement 1E*, *Video 1* and *2*), confirming that a portion of mHtt-94Q molecules can remain in a mobile, dynamic, and soluble state. These results are consistent with previous FRAP and FCS experiments showing fast diffusing and aggregate states of mHtt protein (*Cheng et al., 2013*; *Kim et al., 2002*; *Park et al., 2012*; *Wustner et al., 2012*).

Interestingly, reconstructed molecular density maps from single molecule localization experiments reveal that even Htt-25Q fragments can form small protein polymers or clusters in live cells

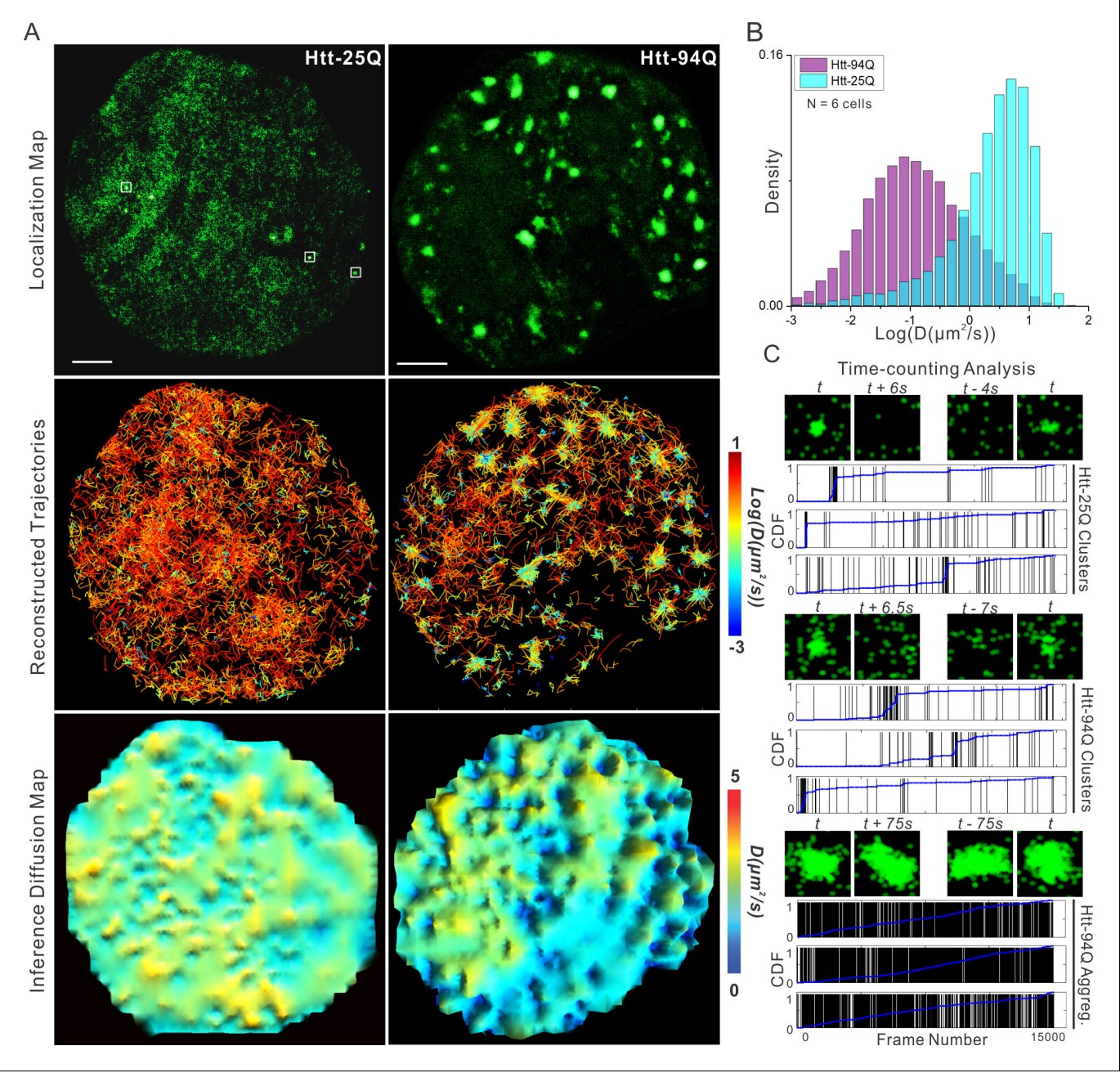

**Figure 1.** mHtt protein displays three distinct states in living cells. (**A**) Top, localization map of live-cell PALM data set for the indicated Htt fragment. The PALM data were recorded at 20 ms per frame using imaging conditions specified in Materials and methods. Diffraction-limited Htt-25Q aggregates are indicated by square boxes (White). Middle, reconstructed single-molecule trajectories reflecting sub-cellular molecular diffusion. Each trajectory is color-colored using the diffusion coefficient calculated by the linear regression of the MSD curve with a $R^2 > 0.9$. 2000 trajectories are shown for the Htt-25Q condition. Twice as many trajectories are shown for the mHtt-94Q condition to ensure relatively equal sampling and representation in regions without mHtt-94Q aggregates. Bottom, diffusion map calculated by a Bayesian inference- based method (**El Beheiry et al., 2015**). Trajectories with minimal 2 connected localizations were considered in this analysis. Each pixel in the image is color-coded with the most probable diffusion coefficient for that pixel calculated by InferenceMap. Scale bars, 2 µm. (**B**) Histogram of diffusion coefficients calculated same as in the middle panel of (**A**) for the indicated condition. Htt-25Q, 7266 trajectories; mHtt-94Q, 19,975 trajectories. N = 6 cells. (**C**) The temporal history of localizations in each selected region: vertical line (Black) indicates the frame from which the localization is detected. The blue trend line represents the cumulative distribution function of the total localizations detected before a given frame number. Three representative regions are shown for each condition. The images above each plot are sliding-window localization maps from **Video 3**, showing the temporal dynamics of Htt protein from each category. *t* indicates when the cluster/aggregate is clearly visible. Scale bars, 2 µm

*Figure 1 continued on next page*

*Figure 1 continued*

The following figure supplement is available for figure 1:

**Figure supplement 1.** Dynamics of Htt protein in live cells.

(*Figure 1A*, white box). These very small clusters are below the diffraction-limit (50~100 nm) and cannot be adequately resolved by conventional imaging methods. Temporal localization history and sliding window analysis (*Figure 1C* and *Figure 1—figure supplement 1D*; *Videos 3* and *4* ) suggest that protein molecules in these clusters display relatively fast association/dissociation dynamics. Specifically, time-counting PALM analysis (tcPALM) (*Cisse et al., 2013*) reveal that these small clusters form in temporal bursts (~10–20 s) (*Figure 1C*), while localization detections in control regions with similar molecular densities show no such apparent bursting behaviors (*Figure 1—figure supplement 1D*). Localization maps for mHtt-94Q cells are more complex. Both small clusters and large aggregates can be detected (*Figure 1A* and *Video 3*). Interestingly, localization detections in small mHtt-94Q clusters display similar bursting kinetics as Htt-25Q clusters (*Figure 1C* and *Video 4*). By contrast, the localizations of large mHtt-94Q aggregates distribute relatively evenly over time, suggesting a much greater structural stability (*Figure 1C*; *Video 3* and *4*). Consistent with this observation, FRAP experiments over longer time-scales found little molecular exchange occurring at large mHtt-94Q aggregates (FRAP recovery lifetime > 45 min, See Materials and methods for calculation details) (*Figure 1—figure supplement 1F*, *Video 1* and *Video 2*). Taken together, our results suggest that there are likely at least three distinct states for Htt proteins in living cells: 1) rapidly diffusing, 2) short-lived dynamic clustering and 3) very stable aggregates formed only by the Gln-expanded mutant protein but not the wild-type Htt. It is likely that PolyQ expansion tilts the kinetic balance of mHtt from a dynamic clustering state to an aggregated state, akin to recently described protein dynamic polymer/phase transitions associated with ALS/FTD neurodegenerative disorders (*Kato et al., 2012*; *Molliex et al., 2015*; *Murakami et al., 2015*; *Nott et al., 2015*; *Patel et al., 2015*; *Xiang et al., 2015*). These results also support previous in vitro super-resolution imaging experiment showing that there are likely distinct Htt aggregate species for seeding and polymerization states (*Duim et al., 2014*; *Sahl et al., 2015*; *2012*).

## mHtt aggregates exclude chromatin

Dysregulation of transcription has been extensively implicated in the pathological processes of HD for decades (*Dunah et al., 2002*; *Kumar et al., 2014*; *Mohan et al., 2014*; *Nucifora et al., 2001*; *Paul et al., 2014*). Although altered chromatin structures have been reported in diseased cells (*Labbadia et al., 2011*), ultra-fine chromatin structure and genome organization have not been extensively investigated. To address this, we labeled chromatin by stably expressing H2B-HaloTag in the mHtt-94Q cells. To avoid biases introduced by individual imaging techniques, we imaged the samples with two-color 3D live-cell structured illumination imaging, Airyscan imaging and single-molecule localization microscopy. All these experiments showed consistent results that intensities of H2B staining were dramatically depleted in regions of mHtt aggregates (*Figure 2A* and *Figure 2—figure supplement 1A*; *Video 5*), suggesting that these 'inclusion bodies' likely exclude chromatin. We note here that the size and shape of mHtt aggregates varies significantly in the cell. Thus, to quantify exclusion or recruitment by mHtt aggregates in an unbiased manner, we used a size-normalized, averaging based analysis method. Briefly, we rescale each mHtt-aggregate containing region to have a diameter of 100 pixels (*Figure 2B*). Then, an intensity map was generated by averaging multiple mHtt aggregate regions obtained across different cells. The center-to-peripheral radial grayscale intensity curve is then calculated by each circular pixel increment

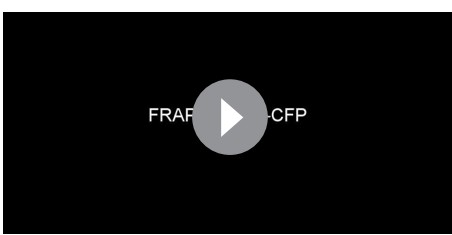

**Video 1.** FRAP experiments in mHtt aggregate negative (left) and positive (right) cells. The time lapse confocal imaging interval for the aggregate negative cell is 0.5 s. That for the aggregate positive cell is 8 s.

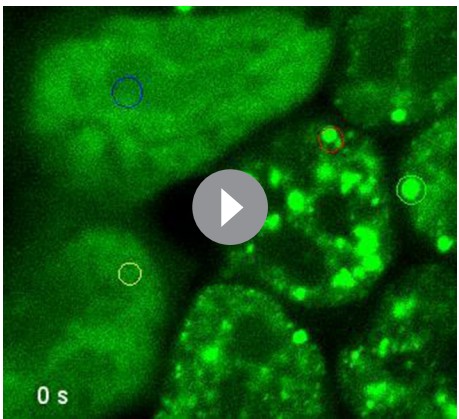

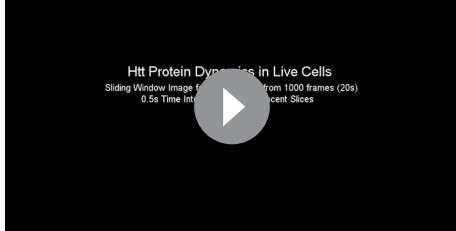

**Video 3.** Sliding-window representation for live-cell PALM datasets of Htt-25Q (Left) and mHtt-94Q (Right). PALM Imaging is performed with a rate of 50 Hz. In each frame, single-molecule localizations from adjacent 1000 frames were pooled and plotted. The interval for the sliding window is 0.5 s (25 frames).

**Video 2.** FRAP experiments in mHtt aggregate negative (left) and positive (right) cells performed in the same field of view, showing that laser power is sufficient for bleaching. The fluorescence recovery in cells with soluble mHtt-94Q is rapid, while that at aggregate regions is much slower.

for both channels (*Figure 2C and D*). For molecular structures (such as H2B) that are excluded by mHtt aggregates, the intensity curve resembles the mirror-image of the curve for for mHtt (*Figure 2D*, *Figure 2—figure supplement 1B and C*). We note that the commonly-used DNA dye DAPI non-specifically stains cytoplasmic mHtt aggregates (*Figure 2—figure supplement 1D*), suggesting that DAPI fluorescence might not be an ideal indicator for genome localization and organization in this case.

## mHtt aggregates impede target search of proteins containing long continuous PolyQ tracts

One intriguing pathological feature of HD is selective cell death in the striatum and certain regions of the cortex (*Ferrante et al., 1985*; *Rikani et al., 2014*). Since Htt is widely expressed in different tissues in the body, it remains largely unclear how mutant Htt causes selective cell death in striatal neuron populations (*Strong et al., 1993*). Interestingly, it was shown that long PolyQ tract containing proteins such as wild-type Htt and TATA box binding protein (TBP) are trapped in the mHtt aggregates (*Kim et al., 2002*; *Lee et al., 2004*). To test whether we could recapitulate these results in our live cell imaging system, we expressed WT Htt-25Q-HaloTag exon 1 fragment in cells with mHtt aggregates. Indeed, Htt-25Q protein becomes significantly sequestered in the mHtt aggregates in both ES cells and STH striatal cells (*Figure 3A* and *Figure 3—figure supplement 1B*). Next, we deleted the 25Q PolyQ tract in the wild-type Htt fragment and found no measurable recruitment of the fragment to mHtt aggregates (*Figure 3B*), suggesting that the 25Q PolyQ tract is likely required for the interaction. As to TBP, we found that the N-terminal PolyQ tract containing domain but not the C-terminal DNA binding domain of TBP is selectively enriched in mHtt aggregates (*Figure 3A–D*), suggesting that recruitment of TBP to mHtt aggregates also requires the PolyQ containing domain. Importantly, these results are consistent with previous studies using different cell lines (*Kim et al., 2002*; *Lee et al., 2004*), suggesting that mechanisms mediating these interactions are likely not cell-type specific.

To identify new mHtt aggregate interacting factors and to test whether the presence of a PolyQ tract is a reliable predictor for recruitment

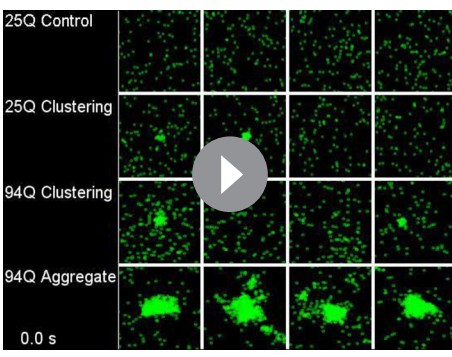

**Video 4.** Zoom-In view of indicated types (Left) of region from *Video 3*, showing distinct local molecular dynamics.

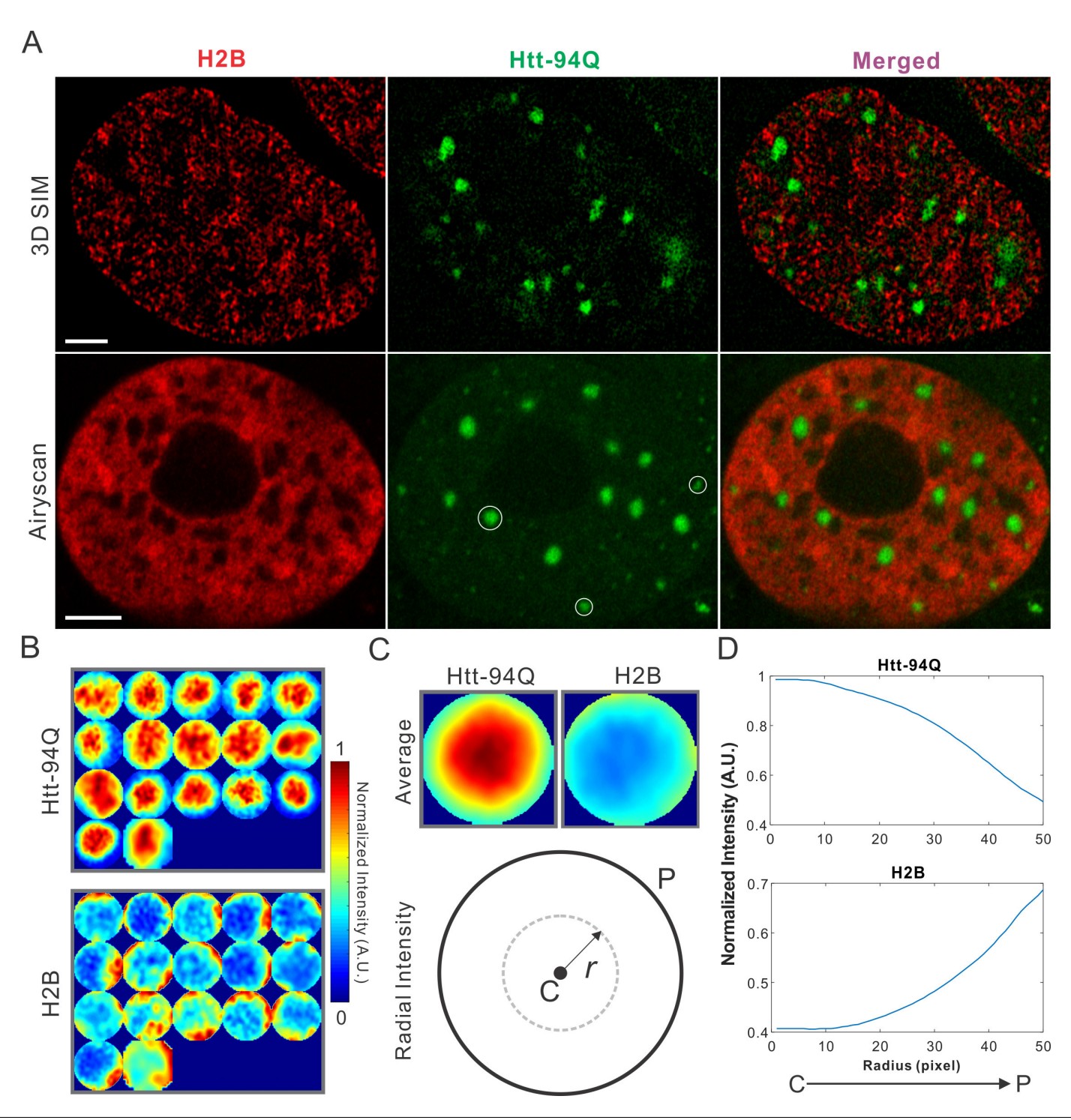

**Figure 2.** mHtt aggregates exclude chromatin. (**A**) Top, live-cell 3D SIM imaging of chromatin (H2B-HaloTag-JF549, red) and mHtt aggregates (mHtt-94Q-CFP, Green). See *Video 5* for 3D rendering. Bottom, Airyscan imaging of fixed cells with the same two-color labeling. (**B**) Size-normalized (100 pixel diameter) fluorescence intensity maps for 17 mHtt aggregate containing regions as circled in (**A**). The intensity maps for corresponding regions in the H2B channel are represented in the bottom panel. (**C**) Intensity maps for different regions shown in (**B**) are summed and averaged for each channel. The mean radial intensity trend line for each channel is calculated as the function of the radius from the center (**C**) to the peripheral (**P**). (**D**) Radial intensities are plotted as the function of the radius for mHtt-94Q (Top) and H2B (Bottom) channel. N, 17 regions from 8 cells. Scale bars, 2 μm

*Figure 2 continued on next page*

*Figure 2 continued*

The following figure supplement is available for figure 2:

**Figure supplement 1.** Map spatial relationship between chromatin and mHtt aggregates.

by mHtt aggregates, we next screened PolyQ containing proteins genome-wide and identified a PolyQ tract containing transcription factor, Foxp2. (*Figure 3—figure supplement 1*). Most interestingly, Foxp2 is a striatum-specific transcription factor shown to be important for speech evolution, motor learning and neural system development (*Enard et al., 2002*; *Groszer et al., 2008*; *Lai et al., 2001*; *Shu et al., 2005*; *Takahashi et al., 2003*). Disruption of both copies of the *Foxp2* gene caused severe motor impairment in mice (*Shu et al., 2005*). Intriguingly, in humanized Foxp2 mice, the medium spiny neurons in the striatum display increased dendrite lengths and increased synaptic plasticity, suggesting that Foxp2 is important for the development of cortico-basal ganglia circuits (*Enard et al., 2009*). These neural circuits are specifically affected in HD (*Albin et al., 1989*; *Ferrante et al., 1985*; *Ross and Tabrizi, 2011*; *Slow et al., 2003*). We found that for the full length Foxp2, the N-terminal PolyQ containing fragment but not the C-terminal DNA binding domain strongly interacts with mHtt aggregates (*Figure 3A and B*, *Figure 3—figure supplement 1B*). Taken together, these results suggest that recruitment of these factors to mHtt aggregates likely involve PolyQ containing domains and thus, cellular functions mediated by these important transcription factors are also likely affected by the presence of mHtt aggregates. Because of the critical role of Foxp2 in HD-affected brain region and function, our findings also pose an intriguing possibility that Foxp2 might be one of the essential effectors responsible for the selective effects of HD on striatal neural circuits (See *DISCUSSION* for details).

## Aromatic amino acids are required for Sp1:mHtt aggregate interactions

Although our recent unbiased screen for long continuous PolyQ tract containing transcription factors did not identify the sparsely Q-rich human TF Sp1, multiple studies had previously implicated the importance of this regulatory protein in HD. Interestingly, early studies showed that over expression of Sp1 and TAF4 (one of its co-activators) can partly rescue mutant Htt induced gene expression defects (*Dunah et al., 2002*), and polyQ mHtt fragments inhibit Sp1-mediated transcriptional activation in vitro (*Zhai et al., 2005*). These results suggested that Sp1 might also interact with mHtt protein aggregates in living cells. To test this directly, we performed live-cell 3D Structured Illumination imaging experiments. Interestingly, we observed substantial Sp1 binding even to cytoplasmic mHtt aggregates (*Figure 4—figure supplement 1A*). Live-cell single-molecule tracking revealed that Sp1 directly binds to mHtt aggregates (*Figure 4A,B* and *Video 6*) and that full length Sp1 is selectively enriched in mHtt aggregate regions (*Figure 4C–E*, and *Figure 4—figure supplement 1B*). These results are somewhat surprising, because although Sp1 is enriched for Q's, there are no obvious continuous PolyQ tracts in the Sp1 protein sequence. To probe the amino acid

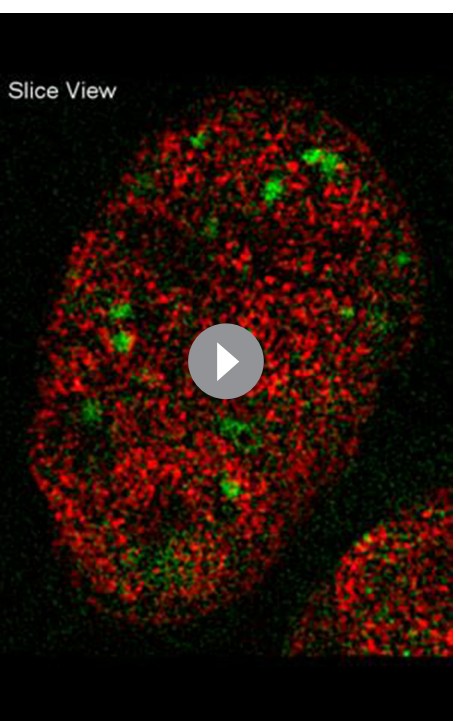

**Video 5.** Live-cell 3D SIM imaging with ES cells expressing H2B-HaloTag (JF549 labeling, Red) and mHtt-94Q-CFP (Green). In the video, 2D slices in the axial direction are looped 3 times followed by a rotational 3D volume rendering.

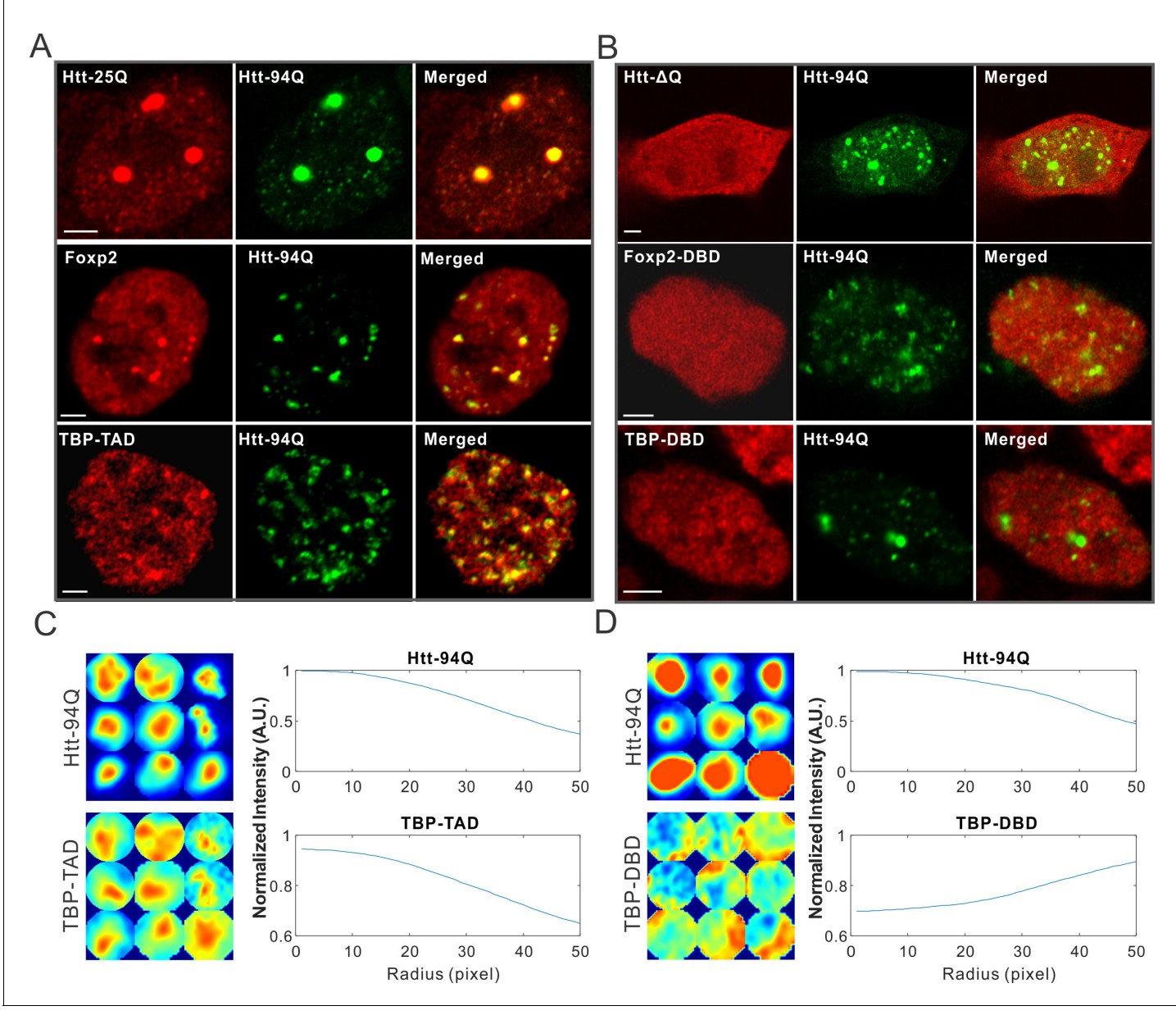

**Figure 3.** mHtt aggregates non-specifically interact with Htt-25Q, Foxp2, and TBP via PolyQ containing domain. (**A**) Airyscan images showing enrichment of Htt-25Q-HaloTag, HaloTag-Foxp2 and HaloTag-TBP-TAD domain (labeled with JF646) at mHtt aggregate regions (mHtt-94Q-mEOS3.2-NLS). (**B**) Deletion of 25Q in Htt-25Q or PolyQ enriched TAD in Foxp1 and TBP abolishes the recruitment of these factors to mHtt aggregates. (**C**) And (**D**) Radial intensity curves as plotted in *Figure 2D* for mHtt-94Q (Top) and TBP-TAD (**C**) or DBD (**D**) (Bottom) channel. N, 9 regions from 5 cells. Scale bars, 2 μm

The following figure supplement is available for figure 3:

**Figure supplement 1.** PolyQ tracts are important for aggregate binding.

sequences of Sp1 responsible for this interaction, we first separated the N-terminal transcription activation domain (TAD) (1–615aa) from the C-terminal DNA binding domain (DBD) (616–781aa). The Sp1 N-terminal TAD showed strong interactions with mHtt aggregates while the C-terminal DBD was excluded from mHtt aggregates (*Figure 4C and D*). To further narrow down the interaction domain, we performed a series of truncation experiments (*Figure 5—figure supplement 1*) and

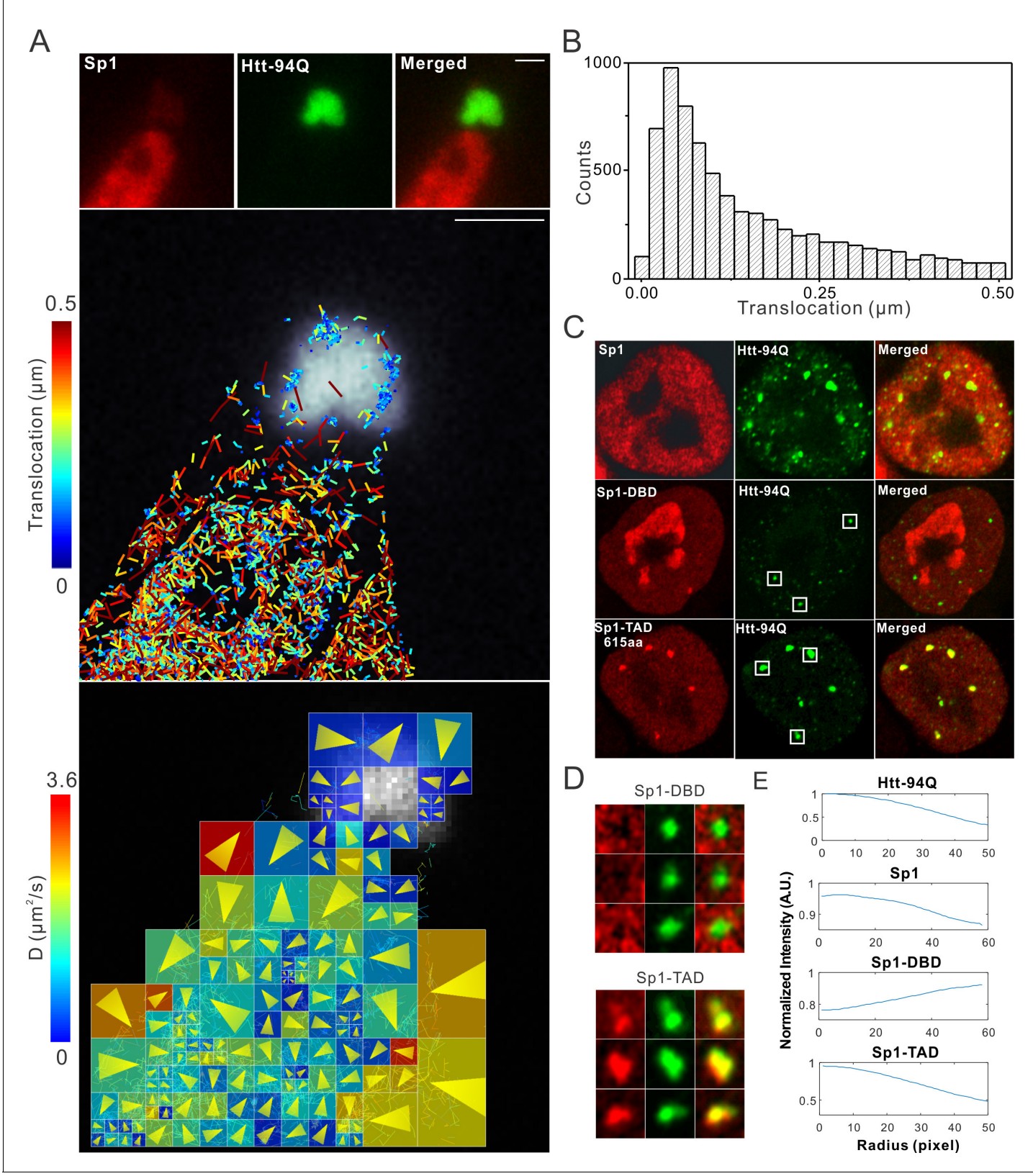

**Figure 4.** Single-molecule imaging of Sp1 binding to mHtt aggregates. (**A**) Single-molecule imaging of cytoplasmic Sp1-mHtt aggregate interaction Top, Epi-fluorescence images of JF549-HaloTag-Sp1 (red) and mHtt-94Q-CFP (green) before single-molecule imaging Middle, 2D molecular interaction map of Sp1 binding to mHtt aggregates. Two-point translocations are color-coded according to the jumping distance shown in (**B**). 7399 translocations
*Figure 4 continued on next page*

*Figure 4 continued*

are included in this analysis. See *Video 6* for single molecule tracks overlaid on raw data. A total of 215 (trajectories) Sp1 binding events were detected on the surface of the aggregate. Bottom, Diffusion map generated by InferenceMap as shown in *Figure 1A*. Each pixel in the image is color-coded with the most probable diffusion coefficient for that pixel. (B) Histogram for jumping distances shown in the middle panel of (A). (C) Two-color Airyscan images of ES cells expressing indicated Sp1 fragment (red, JF646) and mHtt-94Q-mEOS3.2-NLS (green). (D) Zoom-in view of mHtt aggregate regions boxed in (C) shows that Sp1-TAD is recruited to mHtt aggregates while Sp1-DBD is excluded. Radial intensity analysis of HaloTag-Sp1, HaloTag-DBD and HaloTag-TAD (Imaging data are shown in *Figure 4—figure supplement 1B*) shows specific enrichments of Sp1 and Sp1-TAD but depletion of Sp1-DBD at mHtt aggregate regions. Scale bars, 2 μm

The following figure supplement is available for figure 4:

**Figure supplement 1.** Sp1 non-specifically interacts with mHtt aggregates in live cells.

identified a 50aa Sp1 fragment (166–215) with 14Q's that retained the interaction with mHtt (*Figure 5A*). Tethering this fragment to HaloTag allowed us to monitor its selective recruitment to mHtt aggregates (*Figure 5B and D*). Surprisingly, a control fragment from Sp1 (363 - 412aa) also enriched for glutamines (18Q's) failed to interact with mHtt aggregates (*Figure 5C and D*). Sequence gazing revealed that the only apparent difference between these two Q-rich fragments is the presence of three aromatic amino acids interspersed among the 14Qs of Sp1 (166–215aa) (*Figure 5A*). Mutation of these three aromatic amino acids (Y, F, F) to A or even Q severely impaired the ability of the fragment to interact with mHtt aggregates (*Figure 5B*). More interestingly, converting three amino acids in the non-interacting Q-rich Sp1 (363 - 412aa) fragment to phenylalanine (F) animated this fragment to bind mHtt aggregates (*Figure 5C and D*). These results suggest that interspersed aromatic amino acids are likely critical for Sp1 to interact with mHtt aggregates while stretches of PolyQ rich regions are not sufficient and long contiguous PolyQ tracts may not be necessary for mHtt aggregate interactions. Our results indicate that mHtt aggregates display a remarkable degree of sequence or structural specificity as many protein fragments, even some with Q-rich patches fail to interact and become trapped by the large rafts of mHtt protein, consistent with previous studies (*Rajan et al., 2001*). Instead, only certain proteins with specific Q-rich properties such as the presence of interspersed aromatic moieties were found to be retained in the live cell context. Nevertheless, we expect that more complicated and perhaps even some more promiscuous protein features may govern protein:mHtt aggregate interactions. We also anticipate that a broader spectrum of proteins than we have sampled can become trapped and thus affected by mHtt aggregates in different cell-types or cellular contexts. Indeed, previous studies suggest that other transcription factors without long continuous PolyQ stretches such as CBP and TAF4 are likely also sequestered in mHtt aggregates (*Dunah et al., 2002*; *Nucifora et al., 2001*).

## mHtt aggregates waylay cellular target search processes

Complex multi-protein biochemical reactions that drive gene expression and molecular trafficking involve target search processes that are tightly regulated by modulating specific versus non-specific macromolecular interactions within

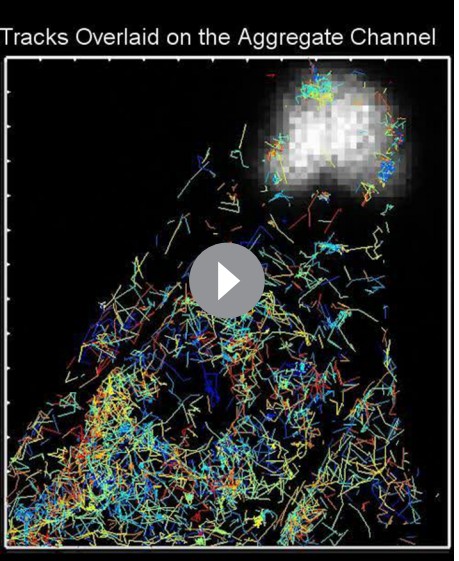

**Video 6.** First segment: Single-molecule trajectories superimposed on Htt-aggregate channel. Trajectories are randomly color-coded. Second segment: Diffusion map calculated by InferenceMap superimposed on Htt-aggregate channel. Third segment: Reconstructed single-molecule trajectories of Sp1 overlaid on raw imaging data with dragon presentation (20 step delayed). Presentation software: InferenceMap

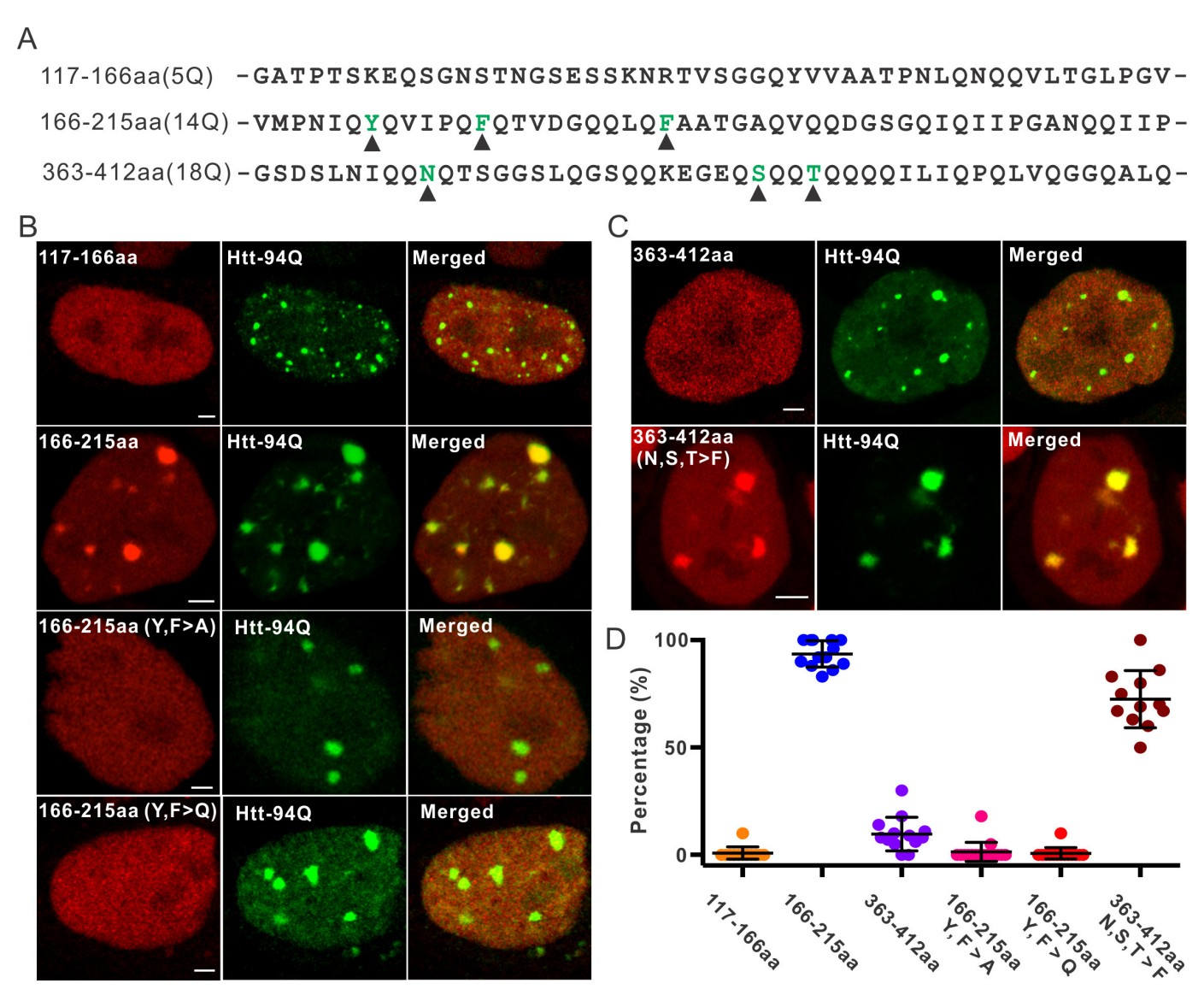

**Figure 5.** Aromatic acids are critical for Sp1 binding to mHtt aggregate. (A) Protein sequence information for Sp1 fragments shown in (B). The number of glutamines in each sequence is in the parenthesis on the left. The amino acids mutated in the panel (B) are highlighted by black triangles below. (B) And (C) – upper panel, Airyscan images show that out of three fragments list in panel (A), only Sp1 166-215aa (14Q) fragment is specifically recruited to mHtt aggregates. Mutation of aromatic amino acids (Y, F) to A or even Q abolishes the interaction. (C) – lower panel, Converting 3 amino acids (N, S, T) in Sp1 363-412aa (18Q) fragment to F enables this fragment to bind mHtt aggregates. (D) Co-localization dot plot for results shown in (B) and (C). Each dot denotes one cell. The percentage of mHtt aggregates displaying enrichment for the indicated Sp1 fragment (X-axis) in single cells is calculated and plotted on the Y-axis. Scale bars, 2 μm

The following figure supplement is available for figure 5:

**Figure supplement 1.** Mapping atypical Sp1:mHtt aggregate interaction domain.

the cell. Previously, we demonstrated that key transcription factor (TF) target search processes in the cell are dominated by non-specific highly transient transactions with chromatin (*Chen et al., 2014b*). One model for HD to affect gene regulation is to imagine that mHtt aggregates form large molecular traps in certain cells. Such a scenario would dramatically increase non-specific trial-and-error sampling by TFs before they reach cognate target sites. To quantitatively evaluate how

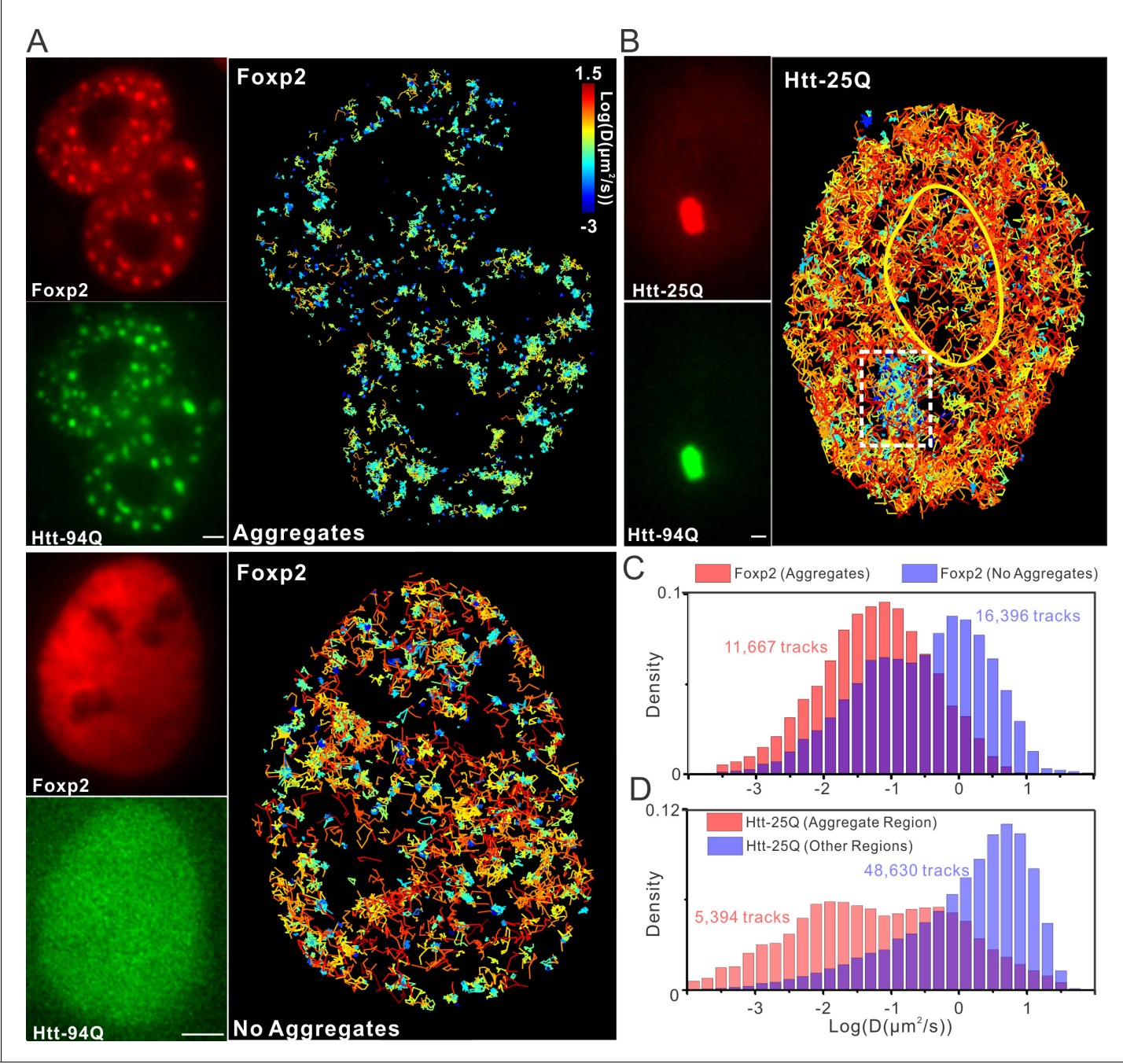

**Figure 6.** mHtt aggregates effectively slow down target search processes in living cells. (**A**) Single-molecule tracking of Foxp2 in mHtt aggregate positive (upper) and negative cells (lower).The resulting single-molecule trajectories are color-coded using the diffusion coefficient calculated by the linear regression of the MSD curve with a $R^2 > 0.9$. 6000 trajectories are shown for the Htt aggregate positive cells (upper). 2000 trajectories for the mHtt aggregate negative cell (lower). The epi-fluorescence images (red, JF646-HaloTag-Foxp2; green, mHtt-94Q-mEOS3.2-NLS) before single molecule imaging are on the left. (**B**) Single-molecule tracking of Htt-25Q in mHtt aggregate containing cells. The color-coding scheme is the same as in (**A**). 2000 trajectories are shown. The epi-fluorescence images (red, JF549-Htt-25Q-HaloTag; green, mHtt-94Q-CFP) before single molecule imaging are on the left. Trajectories are further divided to 'In-aggregate' and 'out-of-aggregate' fractions as shown in *Figure 6—figure supplement 1A and B* for downstream diffusion analysis in (**D**). mHtt aggregate region is indicated by the box with dotted line. The nucleus of the cell is labeled with the yellow contour line. See *Video 7* for reconstructed trajectories overlaid on imaging data. (**C**) and (**D**) Histograms of diffusion coefficients for the indicated condition shown in (**A**) and (**B**). Foxp2 in aggregate positive cells, 11,667 trajectories (6 cells); Foxp2 in aggregate negative cells, 19,975 trajectories (8 cells). Htt-25Q at aggregation regions, 5394 tracks (6 cells); Htt-25Q at non-aggregation regions, 48,630 tracks (6 cells). Scale bars, 2 μm

*Figure 6 continued on next page*

*Figure 6 continued*

The following figure supplement is available for figure 6:

**Figure supplement 1.** mHtt aggregates clog target search.

various parameters (volume, number and size) of mHtt aggregates might affect the target search process, we next performed numerical simulation experiments (See Materials and methods for details). Briefly, we inject a TF into a random position in the nucleus in silico. We assumed that TFs navigate the nucleus via 3D random walk and that the nucleus contains both *bona fide* target sites and mHtt aggregates of various sizes and numbers (*Figure 6—figure supplement 1F* and *Video 9*, See Details in Materials and methods). In the simulation, TF movement is slowed down within the mHtt aggregates as suggested by our SPT experiments (*Figures 4* and *6*). Next, we recorded the average search time for the TF to reach a target site for the first time and found that the search time inversely correlated with the volume ratio of aggregates to the nucleus (VR) (*Figure 6—figure supplement 1C and D*). Strikingly, merely adding aggregates equivalent to ~5% of the total nuclear volume slows down the TF target search process by ~20 fold (*Figure 6—figure supplement 1D*). Interestingly, with a fixed total volume, smaller aggregates more efficiently slow down the target search process, suggesting that merging small aggregates to larger ones might reduce the detrimental effects on target search (*Figure 6—figure supplement 1E*), consistent with the previous report on the protective effects of larger aggregates in cells (*Saudou et al., 1998*; *Taylor et al., 2003*).

To determine whether the target search process of transcription factors and wild-type Htt protein are actually influenced by mHtt aggregates in live cells, we perform SPT experiments on Sp1, Foxp2 and Wt Htt protein. For these three factors, we were able to directly visualize the trapping effects of mHtt aggregates at the single molecule level (*Figures 4*, *6*, *Figure 6—figure supplement 1* and *Figure 7A*; *Video 6*, *7* and *8*). Specifically, diffusion coefficients of these factors become dramatically reduced within mHtt aggregates (*Figure 6A,B*, *Figure 6—figure supplement 1A and B*), indicating binding and longer residence times in these areas. The fraction of slowly diffusing molecules of Sp1, Foxp2 and Wt Htt all become substantially elevated in cells containing mHtt aggregates compared to controls (*Figures 6C,D* and *7A*), consistent with an increased number of non-productive binding events due to greater numbers of decoy non-specific binding sites in the cell. Thus, these direct SMT results support the model that mHtt aggregates form large sticky traps, waylaying the target search process and reducing the cognate target site sampling frequency (*Equations 1–7*).

## Elevated Sp1 levels partially restore the expression of HD affected genes

Reduction of target site sampling or interaction frequencies between TF and cognate DNA target sites would presumably affect downstream biochemical activities such as transcriptional activation or repression (*Chen et al., 2014b*). Here we use Sp1 as the model to test this hypothesis. To do so, we first performed differential gene expression (mRNA-seq) experiments on STH*dh* striatal cell lines derived from Wt (Q7/Q7), heterozygous (Q7/Q111) and homozygous (Q111/Q111) HD mutant mice (*Trettel et al., 2000*). We identified Q111 allele dose-dependent down (449) or up-regulated (558) genes (*Figure 7B and C* and *Supplementary file 2*). To map Sp1 binding sites in the striatal cells, we generated an antigen-purified polyclonal Sp1 antibody for western blot, immuno-precipitation and cell staining (*Figure 7—figure supplement 1B and C*, See Materials and methods for details of antibody generation and purification). This antibody detects a single band of Sp1 of the right size in whole cell extracts and specifically stained the cell nucleus (*Figure 7—figure supplement 1B and C*). Importantly, we confirmed that the antibody is specific because both western blot and nuclear staining of Sp1 disappear when *Sp1*-null ES cell samples were used.

We next performed ChIP-exo experiments with this highly specific anti-Sp1 to map genome-wide Sp1 binding sites in STHdh Q7/Q7 cells (*Trettel et al., 2000*) and identified 6,698 Sp1 target sites in the mouse genome. Most of these binding sites are tightly associated with transcription start sites (TSS) (*Figure 7D*, *Supplementary file 3* and *Supplementary file 4*). Strikingly, Sp1

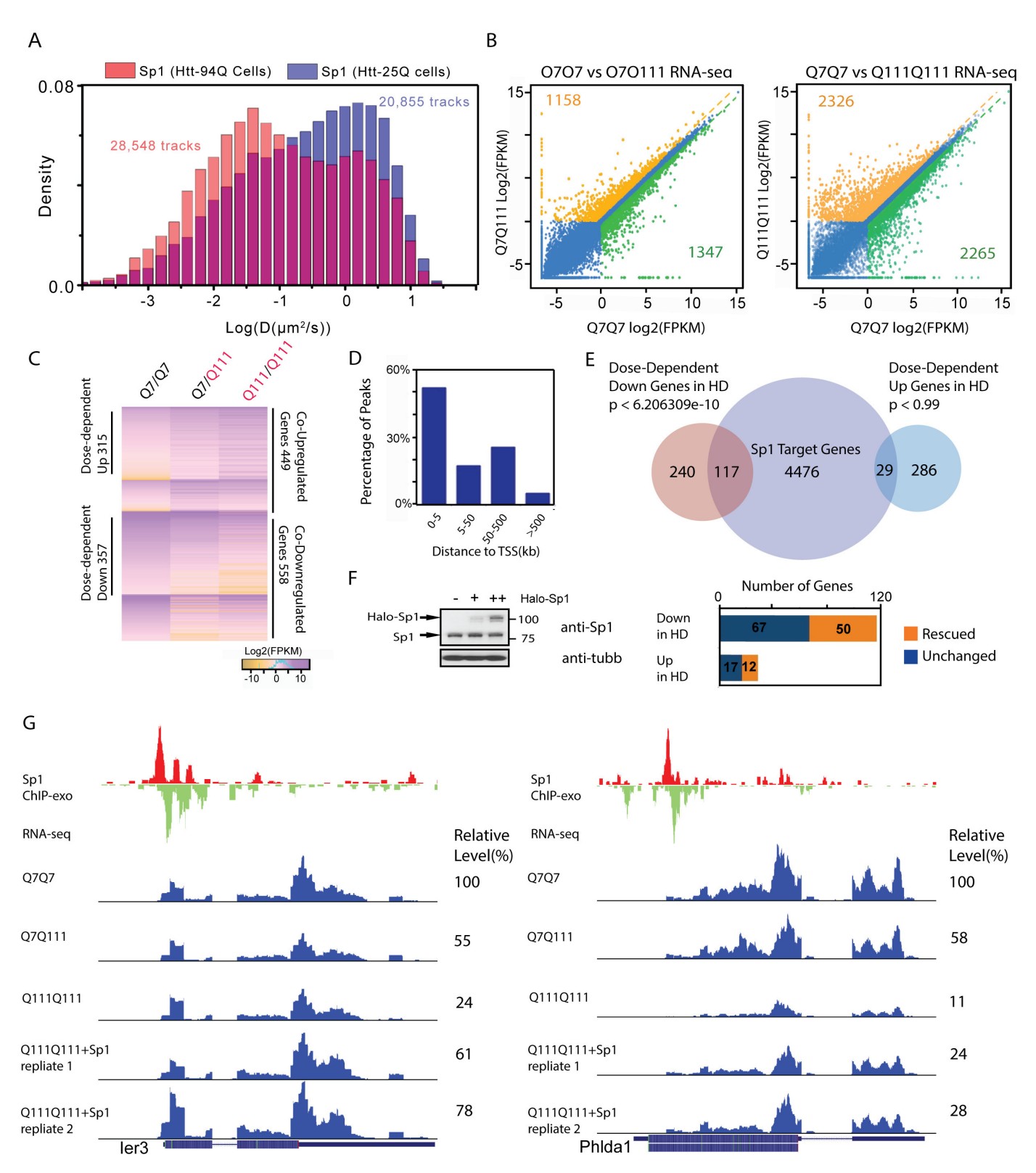

**Figure 7.** Elevated Sp1 levels rescue a subset of Htt-111Q dose-dependent down-regulated genes in cultured striatal cells. (**A**) mHtt aggregates slow down Sp1 diffusion in the nucleus. Histogram (Red) of diffusion coefficients for Sp1 trajectories (28,548) in mHtt aggregate containing cells (N = 12 cells) and that for Sp1 trajectories (20,855) in Htt-25Q control cells (N = 12 cells). In this experiment, HaloTag-Sp1 is labeled with PA-JF646 (See Materials and

*Figure 7 continued on next page*

*Figure 7 continued*

methods for details). (**B**) mRNA-Seq differential gene-expression analysis for indicated sample pairs. Each dot in the plot represents one gene. The log2 scale values of FPKM (Fragments Per Kilobase of transcript per Million mapped reads) of the gene for the comparing samples are plotted. Orange dots, 1.5-fold up-regulation compared to wildtype; green dots, 1.5-fold down compared to wildtype. Orange number, total number of 1.5-fold up-regulated genes; Green number, total number of 1.5-fold down-regulated genes. (**C**) Heatmap showing Q111 dose-dependent gene expression changes. Genes that are up/down regulated in Q111 dose dependent or independent fashion are clustered. Co-regulated up/down genes: genes that are up/down regulated in both Q111/Q7 and Q111/Q111 striatal cells. The expression level of each gene is color-coded. For each group, genes are ranked by the expression levels in the wild-type control (Q7/Q7). (**D**) Percentage of Sp1 ChIP-exo peaks in core and proximal promoter (0–5 kb), distal region (5–50 kb), intergenic region (50–500 kb and > 500 kb) of annotated Refseq TSS in wild type striatal cells (Q7/Q7), the distribution was calculated by GREAT (*McLean et al., 2010*). (**E**) Venn diagram showing overlaps between Sp1 target genes and genes showing Q111 dose-dependent changes in Htt mutant striatal cells. Sp1 target genes are defined as genes with Sp1 peaks within 10 kb of their TSS. 124 of dose-dependent down-regulated genes have Sp1 binding events within 10 kb from their TSS (p<2.07e-8, hypergeometric test). (**F**) Sp1 overexpression in Htt homozygous mutant cells (Q111Q111) rescues the expression levels of a fraction of its direct target genes. Left panel, western blot using anti-Sp1 antibody to show expression of endogenous and Halo-tagged Sp1 in Q111/Q111 striatal cells with no (-), low (+) and high (++) expression level of HaloTag-Sp1. Q111/Q111 lines with high HaloTag-Sp1 expression level were used in RNA-seq experiments to evaluate gene expression rescue. Right panel, Bar graph showing the numbers of rescued (1.5 fold up- or down- regulated compared to Q111/Q111) and unchanged Sp1 dose-dependent targets upon HaloTag-Sp1 overexpression. (**G**) Representative ChIP-exo and RNA-seq tracks of Q111 dose-dependent downregulated Sp1 target genes that were rescued by HaloTag-Sp1 overexpression. Left panel: Ier3, Right panel: Phlda1. ChIP-exo tracks show 5'-end of sequence tags on the sense (red) and anti-sense (green) strands. For RNA-seq tracks, y-axle was set to the same scale (0–460 for Ier3, 0–675 for Phlda1).

The following figure supplement is available for figure 7:

**Figure supplement 1.** Over-expression of Sp1 restores gene expression defects in HD affected striatal cells.

binding sites are significantly enriched around Q111 dose-dependent down-regulated but not up-regulated genes (*Figure 7E*), suggesting that mHtt mostly likely influences Sp1 transcriptional activation of target genes. It's likely that the Sp1 target site sampling frequency is reduced in HD affected cells and this could eventually lead to down regulation of the Sp1 target genes expression. Previously, we showed that tuning-up TF concentrations can increase target site sampling frequencies (*Chen et al., 2014b*) and thus, potentially counteract the Sp1 trapping effects caused by mHtt aggregates. To test this, we over-expressed HaloTag-Sp1 in STH*dh* Q111/Q111 cells (*Figure 7F*) and performed mRNA-seq experiments. A modest elevation (~ 2x) of Sp1 levels partly rescued 50 dose-dependent down-regulated Sp1 target genes, including Rnd1, Phlda1, Ier3 and Myc (*Figure 7F and G*, *Figure 7—figure supplement 1D and E*). Interestingly, Rnd1 is a neuron-specific GTPase important for dendritic spine formation (*Ishikawa et al., 2003*). Both Phlda1 and Ier3 genes are involved in anti-apoptotic pathways (*Neef et al., 2002*; *Wu, 2003*). To further validate whether the Sp1 target sampling frequencies are reduced in Q111/Q111 cells, we performed Sp1 ChIP-qPCR experiments on Ier3 and Rnd1 promoter regions in Q7/Q7 and Q111/Q111 cells. Indeed, the enrichment of Sp1 ChIP-signals were dramatically reduced in Q111/Q111 cells compared to control (*Figure 8A*). Taken together, functional genomic and ChIP studies on the striatal cell culture system further reinforced the concept that mHtt aggregates likely slow down TF target search processes and reduce the target sampling frequencies that can subsequently lead to altered gene expression programs in diseased cells (*Figure 8B–D*).

It is worth noting that consistent with a recent report identifying the Cth (Cystathionine Gamma-Lyase) gene as a key HD disease effector (*Paul et al., 2014*), our genomic assays faithfully

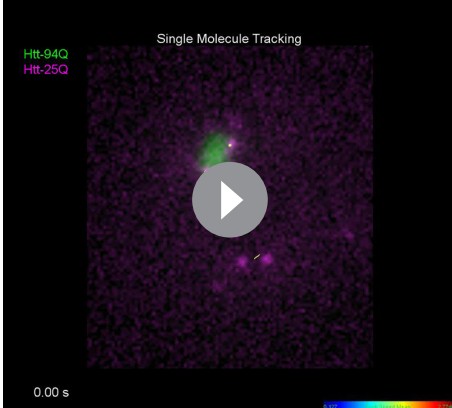

**Video 7.** Reconstructed single-molecule trajectories of Htt-25Q (HaloTag, JF549) overlaid on imaging data in STH*dh* cells with dragon presentation (6 step delayed). Magenta; Htt-25Q single-molecule imaging. Green; mHtt-94Q-CFP Epi-fluorescence image. Presentation software: Imaris.

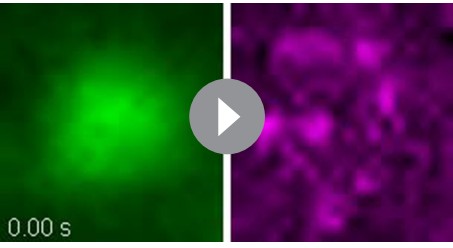

**Video 8.** Zoom-in view at the mHtt aggregate containing region, showing non-specific trapping of Foxp2 at the single-molecule level. Presentation software: Fiji.

detected Cth as a Q111 dose-dependent down-regulated gene (*Figure 7—figure supplement 1A*) and Cth gene expression can be partially rescued by Sp1 overexpression (*Figure 7—figure supplement 1E*). However, we did not detect Sp1 binding at the proximal enhancer region of the gene, suggesting that long-distance enhancer regulation might be involved at this locus.

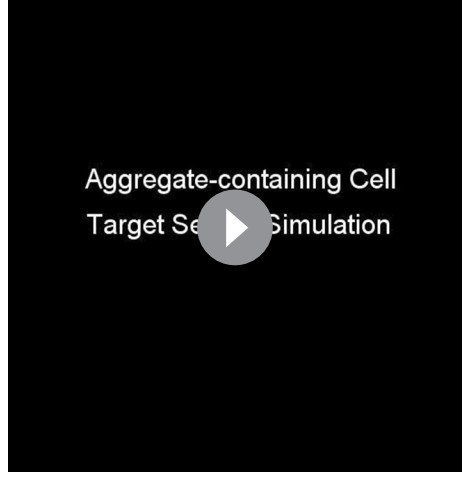

**Video 9.** Target search simulation in Htt-aggregate containing cell. Small black dots and semi-transparent spheres represent target sites and mHtt aggregates, respectively. TF molecule movement is slowed down (red fragments) in the aggregate regions. Presentation software: Matlab

## Discussion

### mHtt aggregates alter molecular kinetics

Using advanced live-cell, single molecule imaging, we were able to directly visualize slower diffusion and retention of key neurological disease related regulatory factors to the mHtt aggregates, effectively operating as large sticky molecular traps. With an abundance of these decoy sites snagging

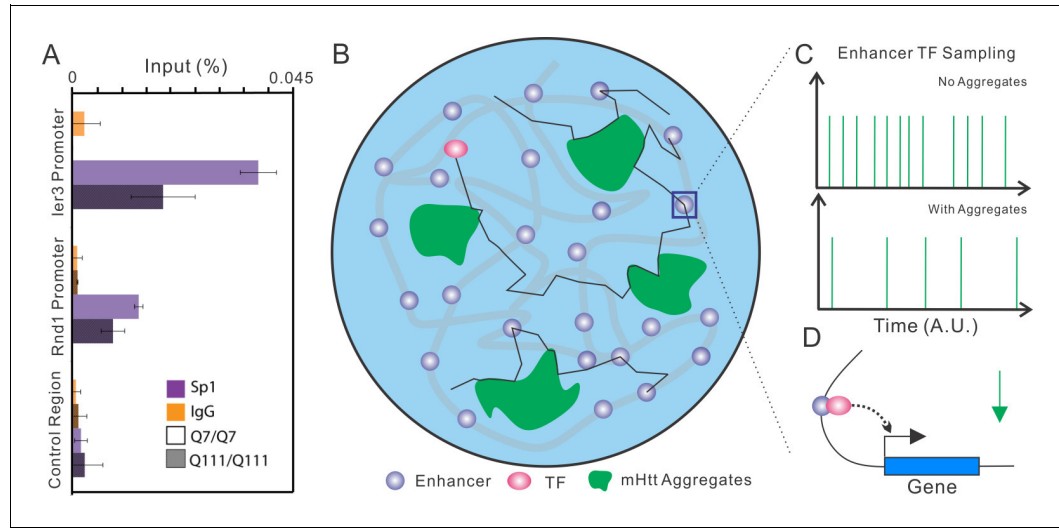

**Figure 8.** Model of Htt aggregates affecting TF dynamics. (A) Sp1 target site occupancies in Q7/Q7 and Q111/Q111 cells assayed by ChIP-qPCR. The ChIP signals are normalized to the amount of input chromatin DNA. (B–D) Large numbers of decoy binding sites created by Htt aggregates in the cell increase trial-and-error collision steps before a TF finds its cognate target site. So, the apparent $K_{on}$ of TF binding to specific sites is reduced. This leads to reduction of target site sampling frequencies (C). Presumably if the TF is a critical transcriptional activator for a gene, the target gene expression will be impeded as well (D).

TFs in the nucleus and the cytoplasm, target searching by transcription factors must undergo many more trials and nonproductive collisions before reaching their designated cognate binding sites to activate the expression of essential genes. Our simulation experiments determined that these large molecular traps greatly increased target search times and reduced target site sampling. In addition, if TF molecules become permanently incorporated into mHtt aggregates, this would effectively lead to the reduction in copy number of actively searching TF molecules in the nucleus. Both mechanisms would inevitably reduce the effective concentration of TFs in the cells and thus impair molecular kinetics associated with these factors (*Equations 1–8*). Consistent with this model, elevating concentrations of TFs such as Sp1 counteracts the Q111 induced gene expression changes in Htt mutant striatal cells. Our SMT experiments suggest that the biochemical reactions associated with Foxp2 and wild type Htt-25Q also become impaired by the same mechanism.

The observation that Foxp2 is recruited to mHtt aggregates via its TAD is particularly interesting, as Foxp2 is a critical factor for the cortical-basal ganglia circuits, which is the most affected brain region in HD (*Enard et al., 2009*). Genetic studies suggest that disruption of Foxp2 function results in speech and language impairment, which is a common HD symptom (*Albin et al., 1989*; *Lai et al., 2001*; *Ross and Tabrizi, 2011*; *Takahashi et al., 2003*). Interestingly, it was also shown that cerebellar defects were observed in Foxp2 null mice, with Purkinje cells particularly affected (*Shu et al., 2005*). Consistent with this observation, extensive studies suggest reduced Purkinje cell density in patients with HD (*Jeste et al., 1984*; *Rees et al., 2014*; *Rub et al., 2013*). Together, our findings strongly suggest that Foxp2 might be one of the essential effectors responsible for the selective disruption of specific neuronal cell populations in patients with HD.

Wild-type Htt has been reported to be involved with axon trafficking in neurons (*Gauthier et al., 2004*; *Strehlow et al., 2007*; *Trushina et al., 2004*). Genetic studies show that knockout of Htt induces certain HD phenotypes and overexpression of wild-type Htt rescues these phenotypes (*Cattaneo et al., 2005*; *Gauthier et al., 2004*), implying that dominant negative effects of mHtt on the wild type protein might contribute to the disease state (*Ho et al., 2001*; *Nasir et al., 1995*; *Zeitlin et al., 1995*). Here, we provided direct evidence to support this model, as the wild-type Htt-25Q fragment also becomes selectively trapped in mHtt aggregates. Together, our data suggest that disrupting the finely tuned molecular kinetics of key regulatory proteins by mHtt aggregates may represent an important common mechanism underlying HD and we speculate that other protein aggregate forming diseases may share a similar disease mechanism.

## Links between transcriptional activation and mhtt aggregates

Early studies of classical enhancer binding proteins suggested that low-complexity peptides such as Q-rich domains are important for transcriptional activation (*Courey et al., 1989*). Detailed mutagenesis experiments revealed that in addition to Qs, hydrophobic amino acids within these Q-rich sequences are also critical for transcriptional activation (*Gill et al., 1994*). Remarkably here, our domain mapping and mutagenesis experiments found that the sequence motif in Sp1 responsible for mHtt aggregate binding is the exact same Sp1 TAD region A required for transcriptional activation. Consistent with our results, recent studies suggest that aromatic residues are essential for the formation of amyloid-like structures by low-complexity domains in RNPs (*Kato et al., 2012*; *Molliex et al., 2015*; *Nott et al., 2015*). Thus, in the future, it would be interesting to study how aromatic amino acids affect protein structure of low-complexity sequences. Do aromatic residues promote formation of beta-sheets and position sparse Qs in the right orientation to interact with Poly-Q aggregates? Another important message derived from this study is that continuous PolyQ tracts are not necessary for mHtt aggregate binding. Likely, a much broader spectrum of proteins with low-complexity sequences than we have studied are affected by the presence of mHtt aggregates in the cell.

## Live-cell mHtt aggregate formation dynamics and low-complexity domain RNP aggregation

How PolyQ expansion affects the dynamic behavior of proteins with low-complexity domains in living cells has been difficult to study. Our live-cell, single-molecule PALM imaging experiments suggest that Htt proteins have at least 3 distinct states under physiological conditions of living cells. Specifically, wild-type Htt fragments display fast diffusing and dynamic clustering states, while mutant Htt

fragments are also able to form more stable large aggregates in the cell. These findings are remarkably reminiscent of recent reports regarding the behavior of the low-complexity domain (LCD) containing RNA granule proteins (such as TDP-43, FUS, hnRNPA1). Most interestingly, low-complexity domains of TDP-43 and FUS are able to form reversible liquid droplet/polymer-like hydrogels in vitro and in live cells (*Kato et al., 2012*; *Molliex et al., 2015*; *Murakami et al., 2015*; *Nott et al., 2015*; *Patel et al., 2015*). As revealed by FRAP experiments, the protein in the droplet states is still quite dynamic (recovery time < 1 min). However, ALS/FTD associated mutations make LCDs of TDP-43 and FUS prone to undergo phase transition to form irreversible hydrogels, impairing the RNP granule function (*Patel et al., 2015*; *Xiang et al., 2015*). Spatiotemporal information-rich live-cell PALM experiments allows us to investigate the temporal dynamics of different Htt states. The dynamic clustering phase of Htt protein has a lifetime of 10~20 s, forming clusters of sub-diffractive sizes. The clusters don't grow further and they quickly disassemble, suggesting that the clustering phase is rapidly reversible and thus analogous to the reversible droplet/hydrogel phase of LCDs in RNPs.

In contrast, mHtt aggregates are much more stable with little dynamic exchange of molecules as seen in our long-term FRAP experiments, suggesting that mHtt aggregates may be more akin to irreversible polymeric-hydrogel states. It seems likely that PolyQ expansion in certain proteins promote the formation of irreversible phase transitions to the aggregated state overcoming the reversible dynamic clustering state. Although the underlying protein sequence requirements of various LCDs such as the Q-rich ones studied here may be quite different from other LCDs, our studies revealed a remarkable resemblance in protein dynamics associated with LCDs in RNPs and PolyQ expansion proteins. Mutations in both classes of LCDs cause severe neurodegenerative diseases. Recent structural studies suggest that beta-sheets are likely structural motifs underlying the formation of liquid droplets and hydrogel polymers for LCDs in RNPs (*Xiang et al., 2015*). It would be interesting in future studies to determine whether similar structural motifs form the basis for PolyQ expansion induced protein aggregation.

# Materials and methods

## Cell culturing

Mouse D3 (ATCC, Manassas, VA) ES cells were maintained on 0.1% gelatin coated plates in the absence of feeder cells. The ES cell medium was prepared by supplementing knockout DMEM (Invitrogen, Carlsbad, CA) with 15% FBS, 1 mM glutamax, 0.1 mM nonessential amino acids, 1 mM sodium pyruvate, 0.1 mM 2-mercaptoethanol and 1000 units of LIF (Millipore). Striatal STHdh Q7/Q7, Q7Q111, Q111/Q111 cells were cultured at 33°C in the high-glucose DMEM medium (without phenol-red) supplemented with 10% FBS, 1 mM glutamax and 1 mM sodium pyruvate. All cell lines were authenticated by genome-wide gene expression profiling (mRNA-seq) and tested as mycoplasma negative.

## Plasmid construction

Sp1, TBP and H2B cDNA was amplified from ES cell cDNA libraries. Foxp2 cDNA was obtained from GE Dharmacon (Cat. #: MMM1013-202798679). Htt-94Q and Htt-25Q cDNA were obtained from Addgene (Plasmid #23966; Htt-94Q and Plasmid #1177; Htt-25Q). Subsequently, full-length and truncated protein fragments were cloned into Piggybac transposon vector (PB533A-2, System Biosciences) or a modified Piggybac transposon vector with PuroR using indicated primers or gBlock fragments (IDT) (*Supplementary file 1*). HaloTag (Promega, Madison, WI) or mEOS3.2 (Addgene: Plasmid #54525) was further cloned to fuse with fragments at the N-terminus (Sp1, Sp1 fragment, TBP) or C-terminus (H2B, Htt-25Q and Htt-94Q). The primer information for cloning is in *Supplementary file 1*.

## Stable cell line generation

Stable cell lines were generated by co-transfection of Piggybac transposon vector with a helper plasmid that over-expresses Piggybac transposase (Supper Piggybac Transposase, System Biosciences). 48 hr post-transfection, ES or STHdh cells were subjected to neomycin or puromycin (Invitrogen, Carlsbad, CA) selection. For electroporation, ES cells or STHdh cells were first dissociated by trypsin into single cells. Transfection was conducted by using the Nucleofector Kits for

Mouse Embryonic Stem Cells (Lonza, Basel, Switzerland) or Nucleofector Kits for Mouse Neural Stem Cells (Lonza, Basel, Switzerland). For the second vector, we use the same piggybac genome integration system. Instead of drug selection, fluorescence activated cell sorting was used to separate stably transfected cells.

## Cell labeling strategy and preparation for imaging

One day before the imaging experiment, ES cells were plated onto an ultra-clean cover glass pre-coated with IMatrix-511 (Clontech, Mountain View, CA). Similarly, striatal cells were plated onto an ultra-clean cover glass without coating. The striatal cell imaging medium was the same as the culturing medium. ES cell imaging experiments were performed in the ES cell imaging medium, which was prepared by supplementing FluoroBrite medium (Invitrogen) with 10% FBS, 1 mM glutamax, 0.1 mM nonessential amino acids, 1 mM sodium pyruvate, 10 mM Hepes (pH 7.2~7.5), 0.1 mM 2-mercaptoethanol and 1000 units of LIF (Millipore).

For single-molecule imaging, we first tested the optimal HaloTag-JF549 and HaloTag-JF647 labeling concentrations. Briefly, chemical structures and synthesis procedures of JF549-HTL and JF646-HTL were described previously (*Grimm et al., 2015*). Several concentrations of JF549-HTL and JF646-HTL (0.5 nM, 1 nM, 2 nM and 5 nM) were used to treat cells for 15 min and then cells were washed with imaging medium for 3 times. The cover glasses were then transferred to live-cell culturing metal holders and mounted onto the microscope one by one. Proper HaloTag-JF549 or HaloTag-JF646 labeling concentrations were determined by the criterion that single-molecules can be easily detected after no or a minimal 2 ~ 5 s pre-bleaching. After fixing the labeling concentration for each cell line, we then proceeded to perform the 2D single-molecule imaging experiments. For PA-JF646 labeling of HaloTag-Sp1, cells were incubated with PA-JF646 with final concentration of ~100 nM for one hour. PA-JF646 stands for a photo-activatable (caged) version of JF646 that would become fluorescent upon low-dose 405 nm light irradiation. The chemical structure and synthesis of PA-JF646 were as previously described (*Lavis et al., 2016*).

For 3D structured illumination imaging and Airyscan imaging, cells were incubated with the JF549-HTL or JF646-HTL with final concentrations around 50 nM for 30 min to ensure that the labeling is close to saturation.

## 3D Structured illumination imaging

After saturated JF549-HTL labeling, Htt-94Q-CFP/H2B-HaloTag or Htt-94Q-CFP/HaloTag-Sp1 ES cells on the cover glasses were placed on a home-build 3D SIM microscope with environment control (37°C). Two color live-cell 3D SIM experiments were performed as previously described (*Fiolka et al., 2012*), except that two color data were acquired on a plane-by-plane rather than a volume-by-volume basis to improve the local registration of the two imaging channels within the moving live cell.

## Airyscan imaging

After saturated JF646-HTL or JF549-HTL labeling, cells on the cover glasses were first fixed with 4% paraformaldehyde for 10 min. The samples were treated with VECTASHIELD antifade mounting medium with DAPI and mounted on a glass slide. The imaging was performed on an inverted Carl Zeiss 880 LSM. Before Airyscan imaging, the beam position on the 32 detector array is calibrated to the center. The final image reconstruction is conducted using the manufacturer provided software.

## Radial intensity analysis

For this analysis, we rescale each mHtt-aggregate containing region to have a diameter of 100 pixels. Then, an intensity map is generated by averaging multiple mHtt aggregate regions obtained across different cells. The center-to-peripheral radial grayscale intensity curve is then calculated by each circular pixel increment for both channels.

## Fluorescence recovery after photobleaching (FRAP)

Htt-94Q-CFP-NLS ES cells were cultured on a clean 25 mm cover glass pre-coated with IMatrix-511. The cover glass is mounted into an Attofluor Cell Chamber. The cells were imaged with an inverted Carl Zeiss 880 LSM with the environmental control system (37°C, 5% $CO_2$). CFP imaging and FRAP

were performed using 440 nm laser with 100% laser power and slowed down scanning speed in the FRAP area. The recovering phase (*Figure 1—figure supplement 1F*) of mHtt aggregate FRAP curve is fitted by single exponential decay model to extract FRAP recovery lifetime.

## Single-molecule imaging

2D single molecule imaging experiments were conducted on a Nikon Eclipse Ti microscope equipped with a 100X Oil-immersion Objective lens (Nikon, N.A. = 1.41), a lumencor light source, two filter wheels (Lambda 10–3, Sutter Instrument, Novato, CA), perfect focusing systems and EMCCD (iXon3, Andor, Belfast, United Kingdom) and Tokai Hit (Japan) environmental control (humidity, 37°C, 5% $CO_2$). Proper emission filters (Semrock, Rochester, New York) were switched in front of the cameras for GFP, JF549 or JF646 emission and a band mirror (405/488/561/633 Bright-Line quad-band bandpass filter, Semrock) was used to reflect the laser into the objective. For tracking the fast diffusion of JF549 labeled molecules, we used a 561-nm laser (MPB Lasertech, Quebec, Canada) of of excitation intensity ~ 800 W cm$^{-2}$ with the acquisition time of 10 ms. For tracking fast diffusion of JF646 labeled molecules, we used a 630-nm laser (Vortran Laser Technology, Inc. Sacramento, CA) of excitation intensity ~ 800 W cm$^{-2}$ with the acquisition time of 10 ms. For single-molecule localization microscopy experiment mapping the spatial relationship of chromatin (H2B-HaloTag-JF549) and mHtt aggregates (Htt-94Q-CFP), we used a SOLA light engine (Lumencor, Beaverton, OR) for imaging mHtt aggregates and a 561-nm laser (MPB Lasertech, Quebec, Canada) of excitation intensity ~1 kW cm$^{-2}$ for single-molecule imaging of H2B-HaloTag-JF549 molcules with an acquisition time of 10 ms (561 nm). Htt-25 Q and Htt-94 Q – mEOS3.2 live-cell sptPALM experiment was performed using the 560-nm laser (MPB Lasertech) of excitation intensity ~ 1000 W cm$^{-2}$ for single-molecule detection and a 405-nm laser (Coherent, Santa Clara, CA) of excitation intensity of 40 W cm$^{-2}$ for photo-switching of mEOS3.2 moiety. It is important to note that the fluorescent proteins used in PALM may display triplet blinking in the millisecond time range, therefore most often filtered out by a detector integration times larger than 10 ms. Here we use a camera integration time of 20 ms to minimize the later counting contribution from blinking. Total ~20000 frames were recorded. ~30 k localized events were used for the final imaging reconstruction. PA-JF646 HaloTag-Sp1 live-cell single-molecule tracking experiment was performed using a 630-nm laser (Vortran Laser Technology, Inc. Sacramento, CA) of excitation intensity ~ 800 W cm$^{-2}$ for single-molecule detection and a 405-nm laser (Coherent, Santa Clara, CA) of excitation intensity of 20 W cm$^{-2}$ for photoactivation of PA-JF646 moiety. The acquisition time is 10 ms. Before single-molecule imaging, we took epi-fluorescence images in all labeling channels for later references. We calibrated our imaging system to achieve minimal drift during acquisition (xy drift <10 nm per hour).

## Single-molecule localization, tracking and diffusion analysis

For single molecule localization and tracking, the spot localization (x,y) was obtained through 2D Gaussian fitting based on MTT algorithms (*Serge et al., 2008*) using home-built Matlab program. The localization and tracking parameters in SPT experiments are listed in the *Supplementary file 1*. MTT algorithm was used to track fast TF dynamics. The resulting tracks were inspected manually. Diffusion coefficients were calculated from tracks with at least 5 consecutive frames by the MSDanalyzer (*Tarantino et al., 2014*) with a minimal fitting $R^2$ of 0.8. For color-coded representation of the trajectory map (*Figures 1*, *4*, *6* and *S6*), each trajectory is color-coded according to its diffusion coefficient with a 'jet' colormap. The trajectory map is reconstructed with the *plot()* function in matlab 2015a. For track division in *Figure 6* and *Figure 6—figure supplement 1*, a binary mask is generated first by using the image from Htt-94Q channel. Trajectories that have partial fragments within the aggregate region are computationally separated for the diffusion analysis. The rest of trajectories are as the control. For sub-regional diffusion analysis in *Figure 6B* and *Figure 6—figure supplement 1A*, binary masks are generated according to mHtt aggregate channel. Trajectories are computationally divided into two populations based on the physical proximity of the trajectory to the mHtt aggregate region. Trajectories contain no localization events in the mHtt aggregate region and trajectories fully or partially included in the mHtt aggregate regions were separated for downstream diffusion analysis.

## Bayesian diffusivity map

Spatial dependence of Htt-25Q (*Figure 1A*), Htt-94Q (*Figure 1A*) and Sp1 (*Figure 4A*) diffusion were analyzed using a Bayesian inference mapping algorithm as previously described (*El Beheiry et al., 2015*). Trajectories were spatially partitioned using a hierarchical (quad-tree) mesh. Dimensions of the zones in this type of mesh were adapted to the characteristic size of the trajectory steps within them, hence accounting for spatially dependent heterogeneities in diffusive behavior. For each zone, the diffusion was presumed to be constant and was calculated by considering all trajectory steps within it (the total length of the trajectory is not consequential). Trajectories were modeled by an overdamped Langevin equation, allowing for physical parameters governing single-molecule movement (e.g. diffusion and interaction energies) to be distinguished. The diffusion coefficient within each zone was calculated as the result of a maximum *a posteriori* estimate from a Bayesian inference calculation.

## Sliding-window and time-counting analysis

The localizations from 1000 frames were pooled and used for reconstruction of a single localization density map. This window slides every 0.5 s (25 frames) until the end of frames. The reconstructed images are temporally ordered and used for the sliding-window video reconstruction (*Video 4*). For time-counting analysis, the localizations from each cluster/aggregate region are selected and the temporal localization history of these localizations are used for the vertical line and the cumulative density plots (*Figure 1C* and *Figure 1—figure supplement 1D*). The control regions were selected based on the criterion that no visible clusters were observed in these regions of the final reconstructed image. The mean lifetime of dynamic clusters in Htt-25Q and Htt-94Q cells are estimated by half of the duration between appearing and disappearing of the cluster in the sliding-window video.

## Translocation analysis

Tracks from the same condition were pooled, and a sliding window of 2 points was applied to each track. The physical distance between two points was calculated by the *pdist2()* function in the Matlab 2015a (MathWorks Inc, Natick, MA). The program iteratively processed all tracks in each category and individual distance were pooled and binned accordingly for the translocation histogram (*Figure 1B*).

## Jumping angle analysis

Tracks from the same condition were pooled, and a sliding window of 3 points was applied to each track. The angle between the vectors of the first two and the last two points was calculated by the *acos()* function in the Matlab 2015a (MathWorks Inc, Natick, MA). The program iteratively processed all tracks in each category and individual angles were pooled and binned accordingly for the angular Rose histogram (*Figure 6—figure supplement 1C*). The minimal jumping distance between two points are set as 40 nm to ensure that the angle measurement is not significantly affected by the localization uncertainty.

## Target search simulation

To initialize the conditions for target search simulation, a sphere with a diameter of 400 units is generated and represents the nucleus of an individual cell. A number of N mHtt aggregates represented by small spheres with diameter of D are randomly allocated within the nucleus sphere. 5,000 TF binding sites (diameter of 2 units) are next randomly allocated within the remaining space, in order to mimic the binding loci within the genome for the transcription factor.

During one simulation, a transcription factor molecule performs 3D Brownian walking unit by unit within the free space of nucleus from a randomly assigned position as previously described (*Liu et al., 2015*), until it arrives at the region of a binding locus. When it walks through the region of mHtt aggregates, the walking speed is reduced by 100-fold. The total number of walking steps is recorded for each simulation. The mean walking step of 2000 simulations is calculated when the values of N, D or VR (volume ratio, total aggregates volume divided by nucleus volume) are varied.

## Antibody production

For antibody used in Sp1 western blot, staining and ChIP experiments, rabbits were immunized with Sp1 residues 1-60aa GST fusion proteins. It is important to note that the sequence of the antigen region is specific to Sp1 but not to other SP family proteins. The antisera obtained were further affinity-purified using MBP-antigen fusion protein immobilized on Affigel 10/15 resin (Bio-Rad, Hercules, CA).

## Western blot

Whole cell extracts from ES Cells and SHT*dh* striatal cells were isolated using RIPA buffer that contained Complete Protease Inhibitor Cocktail (Roche, Basel, Switzerland). Protein concentrations were measured using Bio-Rad Protein Assay against BSA standards (Bio-Rad, Hercules, CA). Protein from each sample was resolved by SDS-PAGE. Primary antibodies used: Sp1 (Custom-made, 1:1000) and beta-tubulin (ab6046, Abcam). HRP conjugated secondary antibodies (Pierce, ThermoFisher Scientific, Waltham, MA) were used at a dilution of 1:5000. Western Lightning Plus–ECL (PerkinElmer, Waltham, MA) was used for chemiluminescent detection.

## Immunofloresence staining

ES cells were first fixed with 4% paraformaldehyde, permeabilized with PBST (PBS plus 0.2% Triton X-100) and blocked with 10% FCS and 1% BSA in PBST. Samples were stained with Sp1 primary antibody (1:100) in blocking solution. Secondary antibodies: DyLight 549 conjugated secondary antibodies (anti-rabbit, 1:400, Jackson ImmunoResearch, West Grove, PA). Nuclei were counterstained with DAPI.

## ChIP-exo library preparation

Chromatin Immunoprecipitation (ChIP) was performed according to (*Boyer et al., 2006*) with minor modifications. Briefly, cross-linked EB chromatin was sheared using Covaris S2 system to a size range of 100bp ~ 400bp. Immuno-precipitation was conducted with Sp1 antibody-conjugated Protein A Sepharose beads (GE Healthcare). ChIP-exo library was prepared by following the published protocol with minor modifications (*Rhee and Pugh, 2011*). Specifically, we adapted the SoLid sequencer adaptors/primers to make the final library compatible with the illumina Tru-seq small-RNA system. The detailed primer information is in *Supplementary file 1*.

## Chip-exo peak calling and bound-region definition

We sequenced exo libraries in 50bp single-end format by using the illumina HiSeq platform. After removal of the 3′ most 14 bp which tend to have higher error rates, we mapped our sequencing data back to the mouse reference genome (mm10) by Bowtie 2 (*Langmead and Salzberg, 2012*). After mapping, bound regions (*Supplementary file 3*) were detected by using MACS2 (*Zhang et al., 2008*). Raw sequencing data were deposited to NCBI GEO with the accession number of GSE84058.

## mRNA-seq library preparation

Total RNA from STHdh Q7/Q7, Q7/Q111, Q111/Q111 and Sp1 overexpressed Q111/Q111 was isolated using RNeasy kit (74106, Qiagen, Valencia, CA). mRNA was then purified using Dynabeads Oligo(dT)$_{25}$ (25–61002, Life Technologies, Carlsbad, CA). RNA-seq library was prepared using ScriptSeq v2 RNA-seq Library Preparation Kit (SSV21106, Illumina, San Diego, CA), and then sequenced using an Illumina Hiseq 2000 sequencing platform. Raw sequencing data were deposited to NCBI GEO with the accession number of GSE84058.

## Expression level estimation and differential expression testing

We sequenced mRNA-seq samples in 50bp single-end format (1 lane HiSeq per sample). Reads were mapped to the mouse mm10 genome using Tophat, and Read counts were tallied for each Ensembl annotated protein-coding gene (Ensembl 61) incremented by 1 and differential expression tested using Cuffdiff using all qualified samples. Cuffdiff results were further analyzed by CummeRbund (*Trapnell et al., 2009*; *Trapnell et al., 2012*).

## Mathematic links between number of non-specific sites and target site sampling frequencies

In order to reveal the relationship between Number of non-specific sites ($N_{ns}$), the search time to the specific site ($\tau_{search}$) and the specific site sampling frequencies ($F_{sampling}$). The following terms are defined during the calculation:

$k_{ns}$: association rate to one non-specific site

$k_s$: association rate to one specific site

$N_s$: number of specific binding sites in a cell.

$N_{ns}$: number of nonspecific binding sites in a cell.

$N_{TF}$: number of transcription factors in a cell.

$P_s^b$: probability for a free particle to bind to a specific site.

$\tau_{ns}$: non-specific residence time.

$\tau_s$: specific residence time.

The probability $P_s^b$ was calculated as follows:

$$P_s^b = \frac{k_s N_s}{k_s N_s + k_{ns} N_{ns}} \tag{1}$$

In particular, $1/P_s^b$ gives the average number of trials ($N_{trials}$) for a TF to reach a specific binding site.

$$N_{trials} = 1 + \frac{k_{ns} N_{ns}}{k_s N_s} \tag{2}$$

The average duration for diffusion between two binding sites

$$\tau_{3D} = \frac{1}{k_s N_s + k_{ns} N_{ns}} \tag{3}$$

Previously, we demonstrate that the total search time for one TF to reach a specific site (*Chen et al., 2014a*)

$$\tau_{search} = N_{trials}(\tau_{3D} + \tau_{ns}) - \tau_{ns} \tag{4}$$

Therefore, by combining (*Equations 2–4*), the specific target search time $\tau_{search}$ can be calculated as

$$\tau_{search} = \frac{1}{k_s N_s} + \tau_{ns} \frac{k_{ns} N_{ns}}{k_s N_s} \tag{5}$$

Based on (*Equation 5*), increasing non-specific binding sites ($N_{ns} \uparrow$) in the cell would directly lead to longer specific search times ($\tau_{search} \uparrow$).

Previously, we also show (*Chen et al., 2014a*) that a first order approximation of specific site sampling interval can be defined as

$$Sampling\ Interval\ (s),\ T_{sampling} = \left(\frac{\tau_{search} + \tau_s}{N_{TF}}\right) N_s \tag{6}$$

Accordingly,

$$Sampling\ frequency,\ F_{sampling} = \frac{1}{T_{sampling}} \tag{7}$$

Based on *Equations 5–7*, increasing $N_{ns} \uparrow$ in the cell would lead to longer specific search times ($\tau_{search} \uparrow$) longer target site sampling intervals ($T_{sampling} \uparrow$) and lower target site sampling frequencies, $F_{sampling} \downarrow$.

Finally, it is important to note that if TF molecules are permanently trapped in mHtt aggregates, this would directly lead to the reduction of concentrations (copy number) of searching TF molecules ($[TF] \downarrow$) in the nucleus. The effective association rates of TF molecules to DNA target sites ($K_{on}^*$) is thus decreased, according to the equation below:

$$K_{on}^* \downarrow = K_{on}[TF] \downarrow \tag{8}$$

This model is not mutually exclusive to the target search model presented above. In fact, both mechanisms might play a role.

## Acknowledgement

We thank M Radcliff, S Moorehead and C Morkunas for assistance.

## Additional information

### Competing interests

RT: Robert Tjian is President of the Howard Hughes Medical Institute (2009-present), one of the three founding funders of eLife, and a member of eLife's Board of Directors. The other authors declare that no competing interests exist.

### Funding

| Funder | Author |
| --- | --- |
| Howard Hughes Medical Institute | Li Li<br>Hui Liu<br>Peng Dong<br>Dong Li<br>Wesley R Legant<br>Jonathan B Grimm<br>Luke D Lavis<br>Eric Betzig<br>Robert Tjian<br>Zhe Liu |

The funders had no role in study design, data collection and interpretation, or the decision to submit the work for publication.

### Author contributions

LL, ZL, Conception and design, Acquisition of data, Analysis and interpretation of data, Drafting or revising the article; HL, PD, Conception and design, Acquisition of data, Analysis and interpretation of data; DL, WRL, EB, Acquisition of data, Analysis and interpretation of data; JBG, LDL, Acquisition of data, Contributed unpublished essential data or reagents; RT, Conception and design, Drafting or revising the article

### Author ORCIDs

Li Li, http://orcid.org/0000-0002-2981-6615
Zhe Liu, http://orcid.org/0000-0002-3592-3150

## Additional files

### Supplementary files

• Supplementary file 1. The table lists primers, gBlocks and localization parameters used in the study.

• Supplementary file 2. RNA-seq analysis of different striatal cell lines (Q7/Q7: wild-type; Q7/Q111: mHtt heterozygous mutant; Q111/Q111 mHtt homozygous mutant; Q111/Q111+Sp1: Sp1 overexpressed mHtt homozygous mutant.

• Supplementary file 3. Sp1 ChIP-exo-seq analysis. Table lists all Sp1 peaks called by MACS2 (Q-value cutoff: 5.00e-02)

• Supplementary file 4. Sp1 target prediction. Table lists genes with Sp1 binding sites within 5 kb from their TSS. Prediction was done by BETA minus

## Major datasets

The following dataset was generated:

| Author(s) | Year | Dataset title | Dataset URL | Database, license, and accessibility information |
| --- | --- | --- | --- | --- |
| Li Li, Zhe Liu | 2016 | Gene Expression Profiling Using Huntington Disease Cell Culture Model and High-resolution Sp1 DNA-binding Site Mapping byChIP-exo in STHdh Q7/Q7 cells | http://www.ncbi.nlm.nih.gov/geo/query/acc.cgi?acc=GSE84058 | Publicly available at NCBI Gene Expression Omnibus (accession no: GSE84058) |

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
