## [Decision Letter]

Thank you for submitting your article "Real-time Imaging of Huntingtin Aggregates Diverting Target Search and Gene Transcription" for consideration by *eLife*. Your article has been reviewed by three peer reviewers, and the evaluation has been overseen by a Reviewing Editor and Jessica Tyler as the Senior Editor. The reviewers have opted to remain anonymous.

The reviewers have discussed the reviews with one another and the Reviewing Editor has drafted this decision to help you prepare a revised submission.

The manuscript by Li et al. is quite nicely done and uses extensive intracellular and molecular imaging to track polyQ proteins/aggregates and associating molecules in cells. Additional experiments add lend quantitative mechanistic insight into the effect of polyQ aggregates on the activity of polyQ-binding TFs. The manuscript represents a very important step in that it moves boldly towards live-cell observations of molecular processes with likely high relevance to Huntington's Disease and possibly other misfolding pathologies. Bringing several methods – notably advanced imaging – to bear on the problem, the authors show data that imply that the more stable of the Huntington aggregates have a role in impeding (slowing down) the target search of a discrete palette of disease-related factors, along with quite a few more intriguing findings. This paper does a nice job of bridging several fields and brings together a diverse array of experimental evidence to support the story.

Although this manuscript contains an impressive amount of different elaborate experiments, it also is a problem that many of the experiments shown do not provide new mechanistic insights, but just add to already published work. The impact of super-resolution imaging is overplayed in places. For example, in Results, "[…]below the diffraction limit (50~100 nm) and cannot be detected by conventional imaging." Super-resolution imaging helps with localization, not detection, which is a function of signal/noise/background levels. Each claim for the need of specialized imaging techniques in the manuscript should be carefully reviewed and described in the text to make it clear to which aspects of the measurements the special techniques contributed. The strongest part of this manuscript is the characterization of the interaction between mutant Htt and Sp1. While the interaction of mutant Htt and Sp1 and its effect on disrupting transcriptional activity are already known (Dunah et al. Science 2002, Li et al. MCB 2002), it is interesting to revisit these findings using more advanced methods. A stronger focus on this aspect and a more in-depth analysis of the data generated here would make this manuscript a stronger candidate for publication.

Overall, barring this reservation, the work is an important contribution and the manuscript can be revised to correct this issue and similarly clarify some experimental issues. Specifically, the authors need to revise the manuscript to address the following points. I note that additional experimentation is not expected; it would seem that the authors have the data at hand to address the technical points raised:

1) The authors need to more clearly identify which findings are entirely novel and which are confirmatory or add quantitative detail about previously studied phenomena. The manuscript should be organized so that initially the authors indicate what is known already (e.g., comparison of their results with the work of Lindquist, Elf, Berg, Kopito, and Moerner and others regarding polyQ interactions) and then they describe how their new methods, especially the super-resolution work adds to that pre-existing knowledge. Two relevant studies that merit mention by Li et al. in the context of their exciting demonstrated trapping effects caused by (nanoscale) mHtt aggregates are:

Rajan et al. Specificity in intracellular protein aggregation and inclusion body formation. Proc. Natl. Acad. Sci. USA 98(23), 13060-13065 (2001).

Sahl et al. Delayed emergence of subdiffraction-sized mutant huntingtin fibrils following inclusion body formation. Q. Rev. Biophys. (2015) doi:10.1017/S0033583515000219.

2) Regarding the biophysical model for TF search and the impact of aggregate-binding. The authors describe this impact in terms of the speed of TF diffusion being "slowed". Particularly since the aggregates are shown to exclude chromatin, it may be more useful to describe the impact of aggregate-binding as reducing the "copy number" of TF molecules participating in search at any given moment. The "copy number" parameterization would have the further benefit of being more consistent with recent modeling efforts, e.g. the work of Johan Elf, Otto Berg, and colleagues.

3) Time-counting PALM analysis: The authors state in the third paragraph of the subsection “Live-cell Htt Dynamics” that localizations in so-called "control regions" (not further specified) are free of temporal bunching events which, wherever observed, are then interpreted to be indications of small cluster formation. The argument would be much strengthened by showing data on the signal of individual mEOS3.2 over time in cells, e.g., on the cytosolic-expressed FP mildly fixed so as to make it immobile. The FP presumably also shows blinking (?), and the timings of single-molecule signals (leading to localizations) are therefore a major factor that may confuse the unambiguous detection of any clustering, it would appear. How are (quite possibly sporadic/rare) photophysics/blinking effects ("an FP just happens to give off many bursts in relatively quick succession", giving more weight to its presence in a particular location) discriminated from true local prolonged dwelling of a molecule? How were the "control regions" selected? What are the bounds/limits of asserting true clustering on the ~10–20 s scale? This section should be expanded.

4) Along somewhat similar lines, the authors should discuss if the argument of apparent diffusion coefficients (as inferred by MSD or Bayesian schemes) being low in the locations of the aggregates (as seen by epifluorescence or reflected in the localization map, Figure 1 top row) is not a circular one. The authors write that "the slow mHtt-94Q species were mainly concentrated around aggregates" (subsection “Live-cell Htt Dynamics”, second paragraph). The trajectories in question stem from mHtt (with photoactivated FP) copies that may well be part of the aggregate itself, i.e. be properly incorporated. What is interpreted as the "trajectory" could just represent static localization precision with additional position noise from any not fully corrected cell (or microscope) motion. The authors should discuss this, and possibly weaken their statements if needed.

5) Some information on how many clear cases of Sp1 binding to large (or smaller) mHtt aggregates were observed (as in Figure 4) would be helpful to feel confident these are not just accidental observations, but specific interactions on the surface of the aggregate. Please add a statement about the frequency of observations of the type shown in Figure 4.

---

## [Author Response]

*Although this manuscript contains an impressive amount of different elaborate experiments, it also is a problem that many of the experiments shown do not provide new mechanistic insights, but just add to already published work. The impact of super-resolution imaging is overplayed in places. For example, in Results, "[…]below the diffraction limit (50~100nm) and cannot be detected by conventional imaging." Super-resolution imaging helps with localization, not detection, which is a function of signal/noise/background levels. Each claim for the need of specialized imaging techniques in the manuscript should be carefully reviewed and described in the text to make it clear to which aspects of the measurements the special techniques contributed. The strongest part of this manuscript is the characterization of the interaction between mutant Htt and Sp1. While the interaction of mutant Htt and Sp1 and its effect on disrupting transcriptional activity are already known (Dunah et al. Science 2002, Li et al. MCB 2002), it is interesting to revisit these findings using more advanced methods. A stronger focus on this aspect and a more in-depth analysis of the data generated here would make this manuscript a stronger candidate for publication.*

We thank reviewers for their thoughtful and constructive comments. Following the reviewers’ suggestions, we updated our manuscript in the Results section to briefly discuss the advantage of each imaging techniques and why we choose to use a specific technique. To better justify the need of live-cell PALM, we change the sentence to "…below the diffraction limit (50~100nm) and thus cannot be not adequately resolved by conventional imaging."

We agree with reviewers that extensive studies have already been performed to characterize huntingtin protein dynamics in vitro and in vivo. In the current study, we aimed to contribute to this field by demonstrating that we can image and analyze the dynamics of huntingtin protein in living cells at the single molecule level. We are excited that some of our results are consistent with previous studies and further reinforce their models with independent methods. These are important control experiments showing the measurements performed by these new imaging tools are accurate and reliable.

Meanwhile, it is important to note that our integrated studies also reveal new insights into Htt aggregate formation and how the mHtt aggregates dynamically affect delicate gene network in live cells. For examples, here are some of our new findings (See more in the Discussion):

1) For the first time, we report the existence of a dynamic clustering state for both wild-type and mutant Htt fragment in live cells and at the same we report and the temporal kinetics of this dynamic clustering (Figure 1).

2) We demonstrate for the first time that Foxp2, a key striatum regulator, is dynamically trapped by mHtt aggregates (Figure 6).

3) Sp1 domain mapping experiments reveal unexpected role of aromatic amino acids in promoting protein:mHtt aggregate interaction (Figure 5).

4) Coupling perturbation with genomic approaches, we systematically characterized how mHtt affects Sp1 mediated transcription programs in neurons. New candidate effector genes have been identified, such as Ier3, Phlda1 and Rnd1 (Figure 7, [Supplementary-material SD4-data]).

*1) The authors need to more clearly identify which findings are entirely novel and which are confirmatory or add quantitative detail about previously studied phenomena. The manuscript should be organized so that initially the authors indicate what is known already (e.g., comparison of their results with the work of Lindquist, Elf, Berg, Kopito, and Moerner and others regarding polyQ interactions) and then they describe how their new methods, especially the super-resolution work adds to that pre-existing knowledge. Two relevant studies that merit mention by Li et al. in the context of their exciting demonstrated trapping effects caused by (nanoscale) mHtt aggregates are:*

Rajan et al. Specificity in intracellular protein aggregation and inclusion body formation. Proc. Natl. Acad. Sci. USA 98(23), 13060-13065 (2001).

*Sahl et al. Delayed emergence of subdiffraction-sized mutant huntingtin fibrils following inclusion body formation. Q. Rev. Biophys. (2015) doi:10.1017/S0033583515000219.*

We agree that it is important to make clear that which of our findings are entirely novel. We have updated our manuscript to address this. Specifically, in each Results section, we first cite previous reports and briefly describe what is known. At the end of the section, we summarize new findings. We thank reviewers for pointing out these two important studies. They are now cited in subsections “Aromatic Amino Acids are Required for Sp1:mHtt Aggregate Interactions” and at the end of the subsection “Live-cell Htt Dynamics” respectively.

*2) Regarding the biophysical model for TF search and the impact of aggregate-binding. The authors describe this impact in terms of the speed of TF diffusion being "slowed". Particularly since the aggregates are shown to exclude chromatin, it may be more useful to describe the impact of aggregate-binding as reducing the "copy number" of TF molecules participating in search at any given moment. The "copy number" parameterization would have the further benefit of being more consistent with recent modeling efforts, e.g. the work of Johan Elf, Otto Berg, and colleagues.*

We agree with reviewers that it is important to discuss both the “copy number” effect and the “target search” model. In the “copy number” model, TF molecules are permanently trapped in mHtt aggregates. This would lead to the reduction of concentrations of searching TF molecules ([TF]↓) in the nucleus. The effective association rate of TF molecules to DNA target sites (Kon*) is thus decreased, according to the equation below:Kon*↓=Kon[TF]↓

In the target search model, the surface of mHtt aggregates create a large number of non-specific sites in the cells. Thus, presumably even when the copy number of available searching TF ([TF]) remains the same, the number of non-specific trial-and-error collision steps before the TF reach a cognate site will be dramatically increased (Kon↓). As a result, the association rate (Kon*) of the TF to specific targets will be decreased, according to the same equation:Kon*↓=Kon↓[TF]

Although the final come out of these two modes of action is the same, it is still important to note that their mechanistic origins are different. In this case, it is likely that both mechanisms play a role. We have emphasized the distinction between these two in the subsection “Mathematic Links between Number of Non-specific sites and Target Site Sampling Frequencies” as well as in the Discussion (the first paragraph).

*3) Time-counting PALM analysis: The authors state in the third paragraph of the subsection “Live-cell Htt Dynamics” that localizations in so-called "control regions" (not further specified) are free of temporal bunching events which, wherever observed, are then interpreted to be indications of small cluster formation. The argument would be much strengthened by showing data on the signal of individual mEOS3.2 over time in cells, e.g., on the cytosolic-expressed FP mildly fixed so as to make it immobile. The FP presumably also shows blinking (?), and the timings of single-molecule signals (leading to localizations) are therefore a major factor that may confuse the unambiguous detection of any clustering, it would appear. How are (quite possibly sporadic/rare) photophysics/blinking effects ("an FP just happens to give off many bursts in relatively quick succession", giving more weight to its presence in a particular location) discriminated from true local prolonged dwelling of a molecule? How were the "control regions" selected? What are the bounds/limits of asserting true clustering on the ~10-20sec scale? This section should be expanded.*

We agree with the reviewers that blinking kinetics of FPs would significantly affect the outcome of the time counting experiment, especially if the fluorophore blinks at similar time scales of imaging acquisition. It is reported that the fluorescent proteins such as mEOS used in PALM may display triplet blinking in the millisecond time range, therefore most often filtered out by a detector integration times larger than 10 ms (Annibale et al., 2011). Here we use a camera integration time of 20 ms to minimize the counting contribution from blinking. On a different note, the control regions were selected based on the criterion that no visible clusters were observed in these regions of the final reconstructed image. We have addressed these questions in detail in the updated Materials and methods.

*4) Along somewhat similar lines, the authors should discuss if the argument of apparent diffusion coefficients (as inferred by MSD or Bayesian schemes) being low in the locations of the aggregates (as seen by epifluorescence or reflected in the localization map, Figure 1 top row) is not a circular one. The authors write that "the slow mHtt-94Q species were mainly concentrated around aggregates" (subsection “Live-cell Htt Dynamics”, second paragraph). The trajectories in question stem from mHtt (with photoactivated FP) copies that may well be part of the aggregate itself, i.e. be properly incorporated. What is interpreted as the "trajectory" could just represent static localization precision with additional position noise from any not fully corrected cell (or microscope) motion. The authors should discuss this, and possibly weaken their statements if needed.*

Previously, we have performed single-molecule tracking with relatively immobile chromatin associated molecules such as H2B (Chen et al., 2014). The diffusion coefficients of H2B molecules are generally < 0.1µm^2^/s, similar with what we observed here for mHtt-94Q in the aggregate region. The slow diffusion is likely contributed from cell motion rather than from microscope drift. Based on beads calibration, the xy drift under our imaging condition is <10nm per hour. We have made this clear in the updated manuscript (subsection “Single-molecule Imaging”).

*5) Some information on how many clear cases of Sp1 binding to large (or smaller) mHtt aggregates were observed (as in Figure 4) would be helpful to feel confident these are not just accidental observations, but specific interactions on the surface of the aggregate. Please add a statement about the frequency of observations of the type shown in Figure 4.*

Thanks for this helpful suggestion. In Figure 4, total 215 (trajectories) Sp1 binding events have been detected on the surface of the aggregate. We have updated the information in the revised manuscript (Figure 4 legend).